# Detection of Pathognomonic Biomarker PrP^Sc^ and the Contribution of Cell Free-Amplification Techniques to the Diagnosis of Prion Diseases

**DOI:** 10.3390/biom10030469

**Published:** 2020-03-19

**Authors:** Hasier Eraña, Jorge M. Charco, Ezequiel González-Miranda, Sandra García-Martínez, Rafael López-Moreno, Miguel A. Pérez-Castro, Carlos M. Díaz-Domínguez, Adrián García-Salvador, Joaquín Castilla

**Affiliations:** 1Center for Cooperative Research in Biosciences (CIC bioGUNE), Basque Research and Technology Alliance (BRTA), Bizkaia Technology Park, Building 801A, 48160 Derio, Spain; herana.atlas@cicbiogune.es (H.E.); jmoreno.atlas@cicbiogune.es (J.M.C.); egmiranda@cicbiogune.es (E.G.-M.); sgarcia.atlas@cicbiogune.es (S.G.-M.); rlopez@cicbiogune.es (R.L.-M.); maperez@cicbiogune.es (M.A.P.-C.); cdiaz@cicbiogune.es (C.M.D.-D.); 2Atlas Molecular Pharma S. L., Bizkaia Technology Park, Building 801A, 48160 Derio, Spain; 3GAIKER Technology Centre, Basque Research and Technology Alliance (BRTA), Bizkaia Technology Park, Building 202, 48170 Zamudio, Spain; garciaad@gaiker.es; 4IKERBASQUE, Basque Foundation for Science, 48013 Bilbao, Spain

**Keywords:** transmissible spongiform encephalopathy, prion disease, PrP^Sc^, diagnostic, PMCA, RT-QuIC

## Abstract

Transmissible spongiform encephalopathies or prion diseases are rapidly progressive neurodegenerative diseases, the clinical manifestation of which can resemble other promptly evolving neurological maladies. Therefore, the unequivocal ante-mortem diagnosis is highly challenging and was only possible by histopathological and immunohistochemical analysis of the brain at necropsy. Although surrogate biomarkers of neurological damage have become invaluable to complement clinical data and provide more accurate diagnostics at early stages, other neurodegenerative diseases show similar alterations hindering the differential diagnosis. To solve that, the detection of the pathognomonic biomarker of disease, PrP^Sc^, the aberrantly folded isoform of the prion protein, could be used. However, the amounts in easily accessible tissues or body fluids at pre-clinical or early clinical stages are extremely low for the standard detection methods. The solution comes from the recent development of in vitro prion propagation techniques, such as Protein Misfolding Cyclic Amplification (PMCA) and Real Time-Quaking Induced Conversion (RT-QuIC), which have been already applied to detect minute amounts of PrP^Sc^ in different matrixes and make early diagnosis of prion diseases feasible in a near future. Herein, the most relevant tissues and body fluids in which PrP^Sc^ has been detected in animals and humans are being reviewed, especially those in which cell-free prion propagation systems have been used with diagnostic purposes.

## 1. Introduction

Transmissible spongiform encephalopathies (TSE) or prion disorders, are rapidly progressive neurodegenerative diseases caused by an aberrantly folded protein that affect several mammalian species, including humans [1]. Diagnosis of these diseases ante-mortem is still challenging for several reasons, despite relevant advances in recent years. First, they show highly varied clinical presentations in humans [2], in part due to the different etiologies of the diseases. They can be sporadic, putatively due to the spontaneous misfolding of prion protein (PrP^C^) to the aberrant isoform (PrP^Sc^) (sporadic Creutzfeldt-Jakob disease (sCJD), sporadic Fatal Insomnia (sFI), and Variably protease-sensitive prionopathy (VPSPr)); genetic, which is associated to mutations in the gene encoding PrP^C^ (*PRNP*) that apparently promote misfolding (genetic Creutzfeldt-Jakob disease (gCJD), Familial Fatal Insomnia (FFI) and Gerstmann-Sträussler-Scheinker syndrome (GSS)); or acquired, in which PrP^Sc^ comes from exogenous sources, such as contaminated meat consumption or surgical procedures (variant Creutzfeldt-Jakob disease (vCJD) and iatrogenic Creutzfeldt-Jakob disease (iCJD)) [3,4]. Each one of these diseases presents with distinct clinical signs and symptoms, especially in the genetic forms of the disease, for which more than 60 genetic alterations have been described [5]. Each of them can appear with distinctive features but also showing striking variability even within individuals of the same family. Moreover, to complicate further the accurate and early diagnosis of prion diseases, some cases of CJD have been reported with unusual clinical presentations [6]. Thus, highly variable manifestations that expand from rapidly progressive dementias that lead to death in less than two years, to slower syndromes that take years to progress, makes early diagnosis on clinical settings highly challenging. Second, the similarity of many of the possible signs observed in prion disorders to those from other neurological conditions such as Alzheimer’s disease, frontotemporal dementia, or certain types of parkinsonism, makes them easy to confuse, hindering differential diagnosis [7]. Other tests used regularly in clinical practice, although helpful, do not completely solve the problems for accurate ante-mortem diagnosis of TSE. Neuroimaging and electroencephalogram results, suggestive of neurodegenerative processes, are not present in many cases of prion disorders or, if present, can be similar to alterations observed in other neurological disorders [8]. Biomarkers detectable in cerebrospinal fluid (CSF) are also routinely used. In fact, an incredible effort has been done in recent years to increase sensitivity and specificity of those assessing neuronal damage such as 14-3-3 protein, total tau, and total/phosphorylated tau ratio, neuron specific enolase (NSE), α-synuclein, S100B or neurofilament (NFL) [9,10]. However, as they are markers of rapid neuronal loss that can be elevated in other rapidly progressive neurodegenerative conditions, the positive predictive value is limited and generally low. Therefore, definite diagnosis of TSE ante-mortem is not achievable by these means, with the exception of genetic prion diseases in which sequencing of *PRNP* gene can certainly support a clinical manifestation-based diagnosis. What is required is the post-mortem confirmation based on neuropathological analysis (spongiosis and astrogliosis are usually the histopathological hallmarks of TSE) and immunohistochemical detection of prion protein aggregates or protease-resistant PrP detection by Western blotting (WB) on distinct brain areas [11,12]. In fact, the misfolded, partially protease-resistant and aggregated isoform of PrP^C^, PrP^Sc^, being the causal agent is the hallmark of TSE and the only pathognomonic biomarker identified so far.

Since Stanley Prusiner proposed the “protein only” hypothesis in 1982, postulating that the causal agent of TSE was an exclusively proteic pathogen composed by aberrantly misfolded PrP^C^, the attention of TSE researchers focused on the study of PrP^Sc^. According to this widely accepted hypothesis, PrP^C^ can misfold through a poorly understood event into PrP^Sc^, which aggregates and becomes insoluble, partially protease-resistant, neurotoxic, and able to induce its aberrant conformation to PrP^C^ through seeding [13,14]. In the case of sporadic or genetic prion disorders the location of the initial misfolding event is unknown, but in the acquired prion diseases, the exogenous PrP^Sc^ makes its way to the central nervous system. Here, PrP^C^ is highly expressed facilitating PrP^Sc^ spreading and its neurotoxic properties lead to neuronal damage. Therefore, formation or acquisition of PrP^Sc^ seeds is the initial event in TSE, which then spreads and multiplies exponentially. This event seems to happen long before the manifestation of the first symptoms of the disease [15]. However, using PrP^Sc^ presence as a biomarker for the ante-mortem diagnosis of TSE has been limited due to several reasons: (1) the amount of PrP^Sc^ at early stages of the disease or before disease onset is too low to be detected by standard biochemical or immunological techniques [WB, immunohistochemistry (IHC) or Enzyme-Linked Immunosorbent Assay (ELISA)], (2) it is unknown how long before the onset of clinical signs does PrP^Sc^ appear and in which tissue or body fluid could be present at initial stages, (3) different TSE, even within the same species, are caused by distinct prion strains (putatively slightly different PrP^Sc^ conformations) with differential pathobiological features that determine the presence in peripheral tissues and body fluids, as well as the time of appearance before disease onset [16,17].

Although PrP^Sc^ can be detected ante-mortem directly from certain tissues in some TSE-affected animals (i.e., rectal biopsies in small ruminants and cervids allow direct detection in lymphoid follicles of the rectal mucosa) [18], in general, PrP^Sc^ amounts in peripheral tissues and body fluids are too low for detection with standard biochemical techniques. However, this limitation for the use of PrP^Sc^ as biomarker has been recently overcome, at least partially, thanks to important advances on cell-free systems for prion propagation [19]. The initial attempts to propagate or amplify PrP^Sc^ in vitro in these systems showed that it was possible to achieve prion replication in a test tube but required high amounts of PrP^Sc^ as seed and long incubation periods. This limitation strongly hindered its usefulness for the detection of minute amounts of prions and thus, for TSE diagnosis. Since this initial breakthrough on prion propagation in vitro in 1994 [13], cell-free systems have been developed further and two methods have emerged as the most effective for the detection of minute amounts of PrP^Sc^. Protein misfolding cyclic amplification (PMCA) came first in 2001, in which originally, a substrate composed by healthy brain homogenates was used as PrP^C^ source to be misfolded by minute amounts of PrP^Sc^ obtained from a diversity of tissues from infected animals. Successive cycles of incubation and sonication, considerably improved the detection limit and accelerated the whole process with respect to the initial systems [20]. Further improvements have been developed during the last two decades, which have allowed to (1) increase its sensitivity (serial PMCA [21] and PMCA with beads [22]), (2) to work with a variety of tissues and body fluids [21,23], (3) work with PrP sources other than brain homogenates [24,25], or (4) adapt the system to the detection of different strains from distinct species [26]. This technique and its variants are widely used in TSE research, because apart from their high sensitivity for the detection of down to 1 attogram of some prion strains [27], the final product obtained from prion propagation replicates the main hallmarks of the original seeds including infectivity [28], strain features (glycosylation pattern and pathobiological characteristics) [29], and reproduces one of the most important phenomena in TSE research, such as the transmission barrier [30]. However, the use of PMCA for diagnostic purposes has been hampered by (1) the need of complicated sonication equipment that makes results variable between laboratories, (2) the need of brain homogenates as substrate, (3) the requirement of serial rounds of PMCA which is time consuming, (4) the high possibility of cross-contamination or spontaneous misfolding of PrP^C^ during serial rounds and (5) the need of a separate readout system such as WB of proteinase-K digested products, which increases the total time required for diagnosis [26]. Although further developments of the technique may improve its value for diagnostic purposes, for instance: the use of recombinant PrP and well-defined environments composed by specific cofactors [24,31,32] that improve consistency due to variability in brain homogenate perfusion quality, or the use of multi-well plate formats that could permit its adaptation to a high-throughput setting [33], or the application of different more automatable readout methods [34]; another cell-free prion propagation system is preferentially used for diagnosis and is already being applied in clinical practice, the real time Quaking induced conversion (RT-QuIC) [35]. This technique is revolutionizing diagnostics of TSE and is being expanded to the detection of other protein misfolding-related neurodegenerative disorders in which protein aggregates with seeding capacity are pathognomonic, such as synucleinopathies and tauopathies [36]. RT-QuIC uses agitation instead of sonication cycles and recombinant protein as PrP^C^ source. Since the first report of Quaking induced conversion of prions [37], it has been further developed. Aiming to diagnosis of TSE, it was adapted to a multi-well plate format and coupled to a readout system based on Thioflavin T (ThT) fluorescence that can be performed by the same equipment in which the reaction is done, being more easily automatable [38]. Thanks to the fluorescence from the amyloid-binding dye, which increases upon aggregate formation, reaction kinetics can be monitored in real-time. An initial lag time of variable duration depending on the seed and substrate used are followed by a fast exponential growth phase. Moreover, reaction conditions have been also finely tuned in order to shorten the process time and to achieve improved sensitivity and specificity, increasing detection limit to the range of femtograms to attograms of seeds [39]. In contrast to PMCA or the Protein Misfolding Shaking Amplification (PMSA) recently derived from PMCA [40], the aggregates produced in RT-QuIC do not mimic faithfully the original seeds, limiting its applications in research although it has been used for drug screening or prion strain typing [35,41]. However, for prion detection has more desirable characteristics than PMCA, mainly regarding standardization, reproducibility, and ease of use, allowing its implementation in clinical practice. In fact, for some prion diseases it is being offered in clinical settings since 2015 in the United States and was included in the diagnostic criteria of the Centre of Disease Control (CDC) in 2018 for sCJD. Similarly, it was also included in European diagnostic criteria used by the UK National CJD Research and Surveillance Unit. However, the gold standard for definitive diagnosis is still based on neuropathology and immunological detection of PrP^Sc^. Despite its good sensitivity and specificity detecting some prion strains in different tissues and biological fluids, such as sCJD seeds in CSF, there is still room for improvement. The main problem delaying its generalized use in clinical practice is that first-generation RT-QuIC, performed using recombinant hamster PrP, was unable to detect some specific prions (i.e., variant CJD, the human prion disease acquired from bovine spongiform encephalopathy contaminated sources; some GSS cases due to specific point mutations; etc.) and sensitivity was quite low for other subtypes [19]. Nonetheless, the use of a new recombinant PrP as substrate [42] has provided a solution to this problem, likely making the detection of PrP^Sc^ in body fluids or tissues by RT-QuIC the next gold standard for ante-mortem diagnosis of TSE.

In order for the combination of PrP^Sc^ (as pathognomonic biomarker) coupled to in vitro prion propagation techniques (allowing the detection of minute amounts) to become a really useful diagnostic tool, there are few factors that require careful consideration: (1) for the different TSE, is PrP^Sc^ present in accessible body fluids or tissues? (2) In which tissues or body fluids, is it present in each case? (3) How long before the onset of clinical manifestations is PrP^Sc^ detectable and in which tissues or body fluids appears first? (4) Do relative PrP^Sc^ amounts in different tissues or body fluids correlate with disease progression? In other words, could this biomarker have a prognostic value?

In this review, we will try to shed some light on these issues gathering the evidences of the presence of PrP^Sc^ in different body fluids and tissues in different TSE, both in humans and other mammals. Those tissues and body fluids where PrP^Sc^ has been detected and could be more useful for ante-mortem diagnosis of TSE will be reviewed and the data obtained from in vitro prion propagation assays highlighted.

## 2. Detection of PrP^Sc^ in Brain

One of the most obvious places where PrP^Sc^ can be found is the central nervous system (CNS) and specifically the brain, where the aggregates are more abundant in the final stage of the disease [43]. In fact, according to the present diagnostic criteria followed in most countries, immunohistochemical or biochemical detection of PrP^Sc^ in brain is required for a definitive diagnosis of TSE in humans [44]. However, due to the invasiveness of such procedure and the reduced benefits that supposes a confirmation of prion disease given the lack of treatments, it is rarely performed ante-mortem [45]. Moreover, the differential distribution of PrP^Sc^ in specific brain areas depending on the particular prion disease [46] hinders the utility of ante-mortem brain biopsies, which could easily provide false negative results if the wrong area is biopsied.

### 2.1. PrP^Sc^ in Brain in Animal Prion Diseases

Regarding the time course of PrP^Sc^ deposition in CNS during the pre-symptomatic stage of the disease, it has been thoroughly studied in animal models of acquired prion disease. Thus, the time course of PrP^Sc^ spreading from peripheral tissues to the CNS has been characterized in several models of infection [47,48,49,50,51,52,53,54] and even distinctive deposition patterns were found after injection of the same prions through different routes [55,56,57]. However, there are few reports trying to unravel the time course of PrP^Sc^ deposition in sporadic and genetic forms of the disease, which are the vast majority in humans. This is mainly because: i. it is impossible to predict the development of a sporadic prion disease, hindering the study of the preclinical stage, ii. the rapidly progressive nature of these diseases and their low prevalence makes it impossible to gather enough cases that are not in the final stage, and iii. the animal models available for sporadic or genetic prion disease are scarce and, except for few examples, require infection to reproduce the prion disease for which they were designed [58]. Therefore, for acquired prion diseases and despite the phenotypic variability observed as a result of distinct administration routes, for the most socioeconomically relevant prion strains (scrapie, bovine spongiform encephalopathy (BSE), Chronic Wasting Disease (CWD), and vCJD), the initial sites of PrP^Sc^ replication in the CNS are known as well as the kinetics of PrP^Sc^ accumulation in the brain and in peripheral tissues in experimental and natural infections [15]. However, despite the brain areas in which early PrP^Sc^ deposition is more probable in acquired animal TSEs is known, brain biopsies are not used for early diagnosis given the presence of PrP^Sc^ in early stages in peripheral tissues, such as lymph nodes, which are much more accessible [18].

### 2.2. PrP^Sc^ in Brain in Human Prion Diseases

In the case of human prion diseases, in which acquired diseases account for less than 1% of the cases, PrP^Sc^ deposition patterns for the different TSE at the final stage of the disease are very well characterized and are actually used to distinguish different forms of the disease and even subtypes or strains [59,60]. However, the great variability that characterizes prion diseases also impacts the interpretation of such neuropathological profiling with unusual distributions or exceptions reported [6]. Despite the difficulty of defining PrP^Sc^ deposition kinetics on sporadic or genetic human prion diseases, some efforts have been directed towards defining the staging of the cerebral pathology in sCJD cases. However, PrP^Sc^ presence at each stage could not be studied in detail [61,62]. Overall, although post-mortem neuropathological examination of the brain and PrP^Sc^ detection is the gold standard criteria for definitive diagnosis, detection of PrP^Sc^ on brain ante-mortem through brain biopsy is controverted. On the one hand, it is not advised due to its invasiveness, its low diagnostic success and the low benefits of a confirmatory diagnosis for untreatable diseases [44]. On the other hand, since it may help to detect other causes of rapidly progressive dementia that are treatable and reversible, in certain clinical cases it may be of utmost importance [45]. The conclusions that can be drawn from the few studies reported on brain biopsies of suspected CJD cases are that: (1) although possibly helpful, all the other diagnostic criteria should come first and if a probable diagnosis is still impossible, a brain biopsy could be performed when some other treatable cause of rapid dementia is suspected. (2) The high heterogeneity of human TSE should be taken into account in order to increase the possibility of successful diagnosis and for that, the participation of all implicated departments is recommended to achieve correct sampling (neurology, pathology, neuroimaging and neurosurgery departments). (3) In light of recent advances on PrP^Sc^ detection by in vitro prion propagation techniques from more accessible tissues or body fluids, as will be reviewed in subsequent sections, brain biopsy should be the last option.

Nevertheless, even though PrP^Sc^ detection in brain tissue is not ideal for ante-mortem diagnosis of TSE neither for humans nor for other mammals, it has been extensively used to develop in vitro prion detection methods such as PMCA or RT-QuIC given the high amount of PrP^Sc^ in comparison to other tissues [42,63,64,65,66,67,68].

## 3. Detection of PrP^Sc^ in Cerebrospinal Fluid

Being in direct contact with the CNS and easily accessible via lumbar puncture, the cerebrospinal fluid has become the preferred site for early detection of PrP^Sc^, which has led to the adaptation of in vitro prion propagation techniques to this body fluid. As mentioned above, the CSF is being used extensively in TSE diagnostic for the detection of surrogate biomarkers of neuronal damage that includes an increasing number of proteins, the levels of which are altered in prion disorders, and that have been extensively reviewed elsewhere [9,69,70,71,72]. However, the change in the levels of many of such surrogate markers are also detected in other neurodegenerative diseases, which complicates the differential diagnosis and requires a protein profiling and checking a panel of the relative levels of multiple biomarkers [71]. Although this review intends to focus on PrP^Sc^ as a biomarker, it is worth to pay some attention to total PrP levels in CSF, which have diagnostic and prognostic value for prion diseases. In fact, total PrP levels were reported to be reduced in CSF of patients suffering from different neurodegenerative disorders, including CJD [73]. Although it could apparently have limited utility in differential diagnosis with other neurodegenerative disorders, further studies using larger cohorts of CJD-affected patients concluded that, in combination with other surrogate biomarkers (i.e., tau and Aβ_42_), total PrP levels are useful to distinguish them from patients suffering from Alzheimer’s disease [74,75]. Apart from sCJD cases included in the previous reports, a recent study analyzed total PrP levels in CSF of iatrogenic CJD cases and several genetic prion diseases, showing also decreased levels in all tested cases except for the GSS associated with P102L mutation [76]. Furthermore, they observed that in sCJD patients at different stages and in the case of asymptomatic carriers of FFI-causing mutation, total PrP is reduced as disease progresses, making it a possible prognostic biomarker and an adequate parameter for evaluating future therapeutic interventions. However, since it is a new approach still in development, during the last year a methodological improvement was presented using mass-spectrometry, instead of ELISA, to detect total PrP in CSF more accurately [77]. Similarly, alternative detection of PrP fragments or truncated forms is being explored what coupled also to mass spectrometry could increase the usefulness of measuring total PrP in CSF for diagnostic and prognostic purposes [78].

### 3.1. PrP^Sc^ in CSF in Animal Prion Diseases

It was long ago proven that CSF from TSE-affected animals was infectious when inoculated to other animal models, demonstrating the presence of PrP^Sc^ seeds [79,80,81]. However, direct detection of prions in CSF by WB or ELISA is not possible due to insufficient sensitivity [82] and require more sophisticated methods, such as the use of fluorescent probes [34] or specific precipitants [83]. Since the development of in vitro prion propagation methods, other alternatives have been abandoned in favor of these more sensitive and specific approaches, and both PMCA and RT-QuIC have been used for PrP^Sc^ detection in CSF in animal prion diseases. PMCA has been applied to the detection of prions in CSF of hamsters infected with scrapie-derived prions, reaching detection limits of approximately 50 attograms PrP^Sc^ [31]. Similarly, but increasing the sensitivity by the use of specific cofactors during the propagation in vitro, CSF from macaques infected with classical BSE and atypical L-type BSE (L-BSE) at the pre-symptomatic stage was also used [84,85]. During the last decade, PMCA has been progressively substituted by the RT-QuIC, for the detection of PrP^Sc^ in CSF of animals affected by prion diseases. This technique has allowed the detection of prions in CSF from cattle affected by BSE type H and L, but not in those affected by classical BSE, in which PrP^Sc^ had been previously detected in RT-QuIC in brain tissues [86]. Furthermore, the RT-QuIC using different recombinant substrates permitted the discrimination of the different BSE strains attending to their reaction kinetics [67]. Like in the case of atypical BSE cases, in sheep naturally affected by scrapie, PrP^Sc^ could be detected in CSF in pre-symptomatic and in symptomatic stages of the disease [87]. On the contrary, although atypical scrapie could be detected in brain tissue [42], no report has been found on atypical scrapie detection in CSF. An improved version of RT-QuIC, also known as second generation RT-QuIC, has been similarly applied to CSF from classical BSE and L-BSE infected goat, in which PrP^Sc^ was detected in preclinical and clinical stage [88]. Finally, it is worth mentioning the detection of Chronic Wasting Disease (CWD) prions in CSF of affected cervids, for which both PMCA and RT-QuIC have been used. In the case of CWD-exposed white-tailed deer, both systems were compared showing similar performance regarding sensitivity (around 50%) and specificity (around 90%) [89]. For the analysis of CWD-infected elk CSF only PMCA was used, showing a generally low sensitivity and being able to detect PrP^Sc^ only in the most advanced stages of disease [90]. Therefore, despite PrP^Sc^ detection in CSF using in vitro propagation systems is feasible for detection of the most common prion diseases in animals (classical scrapie, BSE, and CWD), the presence of infectivity in lymphoid nodes and associated mucosa, which are easier to obtain than CSF, has rendered detection in this body fluid uncommon.

### 3.2. PrP^Sc^ in CSF in Human Prion Diseases

In comparison, for diagnosis of human prion diseases the use of CSF samples to seed in vitro propagation reactions has been explored much more profoundly. Lumbar puncture and extraction of CSF is a usual procedure in patients showing neurological signs and especially in those presenting rapidly progressive dementias. Since the quantification of 14-3-3 protein levels is one of the standardized diagnostic criteria [91], this body fluid is available for most of the patients suspected of a TSE. Since the first human CSF-adapted RT-QuIC was reported in 2011, in which from 48 CSF samples of sCJD, iCJD and from other neurodegenerative pathologies they achieved about 80% sensitivity and 100% specificity [92], improvements mainly in the recombinant PrP used as substrate have allowed to increase the sensitivity and to screen for other TSEs apart from sCJD and iCJD [42,93]. Several studies have been performed with the CSF of different cohorts of patients suffering from distinct prion diseases and using slightly different RT-QuIC procedures (summarized in Table 1).

Overall, in the light of all the studies performed until now, analysis of CSF by RT-QuIC has become one of the most helpful tests to diagnose prion diseases ante-mortem in humans. Although some genetic TSE cannot be detected yet using this approach, the development of new substrates and modifications in the methodology might solve these problems soon [42,106]. In fact, RT-QuIC based on CSF samples is being adapted to other protein-misfolding based neurodegenerative disorders such as some synucleinopathies and tauopathies [36]. Given the success of the RT-QuIC, the almost full specificity and high sensitivity achieved and its ease of use in clinical settings, it has surpassed other methods, such as PMCA for the detection of PrP^Sc^ in CSF. However, there are applications for which CSF analysis by PMCA is preferentially used to answer specific questions that could not be addressed by RT-QuIC. Particularly, PMCA based on transgenic mouse brain homogenate has been used to detect the first human vCJD case heterozygous valine/methionine at codon 129, which manifested clinically as a sCJD. It could be definitively distinguished from a sporadic case thanks to a specific PMCA that selectively propagates vCJD prions while being unable to propagate sCJD or gCJD [107]. Another study using a modified version of PMCA also shows the ability of this technique to detect prions in CSF of vCJD-affected patients with 100% sensitivity and specificity [108].

Although RT-QuIC based on CSF has been proven as the most useful technique for diagnosis of human prion diseases ante-mortem, there are still few issues that need to be addressed for its generalized application in clinical practice. The increasing but still small number of laboratories using this method at present for diagnosis, in many cases depend on external supply of recombinant protein to be used as substrate, that among other technical variations may cause low reproducibility between different labs. The inter-laboratory reproducibility using different protein suppliers and distinct batches has been recently addressed showing good results in a small cohort of CSF samples analyzed by 11 different laboratories worldwide [109], but needs to be further assessed [9]. Finally, another problem that affects not only RT-QuIC, but also any other method for prion detection in CSF, is the ignorance of how long, prior to the onset of clinical signs, is PrP^Sc^ detectable in this body fluid, which will be of utmost importance for the pre-clinical detection of the disease in genetic cases when a therapy is finally available.

## 4. Detection of PrP^Sc^ in the Lymphoreticular System

The involvement of the lymphoreticular system in prion diseases in animals and, more specifically, the presence of prion infectivity and replication in spleen has been known for a long time. Initial reports describe the differential accumulation of prion infectivity in spleen of mouse models with different PrP sequences [110] and the early replication of prions in the spleen being relevant mainly after peripheral inoculations [111,112]. In addition, presence in the spleen after intracerebral inoculation of scrapie prions in mouse and sheep have been also reported, pointing to the lymphoreticular system as an early prion replication site before neuroinvasion [110,113,114,115]. However, it was also noticed early on that the lymphoreticular system was not the only possible route for prions to reach the CNS [116]. Since then, the ability of prions to replicate in lymphoid tissues and their role on systemic prion propagation in the case of peripheral infections, especially in oral infections, has been extensively investigated [117,118]. In short, after oral exposure in the case of naturally transmitted scrapie, CWD, BSE, and vCJD, prions arrive to the small intestine and accumulate in gut-associated lymphoid tissues (GALT), such as Peyer’s patches or other lymphoid organs such as tonsil. From there they spread to other secondary lymphoid organs such as mesenteric nodes and the spleen and, after replication in these tissues, mostly in the follicular dendritic cells (FDC) within the GALT, prions invade the peripheral nervous system. Here, prions can spread from the gastrointestinal tract, GALT or secondary lymphoid organs to the CNS via two different neuroanatomic pathways [119]. However, this neuroinvasion route can be bypassed in the case of experimental infections by intracerebral inoculation or by directly injecting prions in peripheral nervous tissues [112,116]. For other orally acquired prions, this pathway is not as clear as in the case of BSE. In cattle orally challenged with this prion strain, infectivity in guts increases at the beginning (~6 months post-infection) but rapidly decreases, only to be recovered at the final stages of disease, probably by retrograde transport: this suggests that Peyer’s patches can likely act as a prion replication sites for BSE [49,120,121]. Therefore, different prion strains might have distinct capacities to propagate in lymphoid tissues, termed lymphotropism [122], meaning that detection of prions in lymphoreticular system is restricted to some prion diseases and for certain infection routes. This was undoubtedly proved using two different strains of Transmissible Mink Encephalopathy (TME) in hamsters, in which one of the strains was not detected in lymphoid tissue, apparently invading the CNS through the cranial nerves innervating the tongue, while the other was found all over the lymphoreticular system [123]. Anyway, the early multiplication of prions in these tissues for some naturally acquired infections such as scrapie, which are usually due to ingestion of prion contaminated material, aroused the interest of researchers trying to develop ante-mortem detection methods for TSE diagnosis in accessible body fluids and tissues.

### 4.1. PrP^Sc^ in Lymphoreticular System in Animal Prion Diseases

Direct detection of protease-resistant protein in the lymphoreticular system of scrapie-affected sheep by WB or IHC was demonstrated to be possible in the pre-clinical stage (in 14 months after inoculation) as well as in sheep showing clinical signs within spleen and lymph nodes, albeit with low sensitivity with respect to brain tissue in the latter ones [124]. Other similar direct-detection studies reported detection of PrP^Sc^ in 87% of the spleen and lymph nodes analyzed in comparison to 100% in brain [125] and in 100% of spleens but in 80% of lymph nodes [126]. Initial attempts to improve sensitivity included improved protocols for PrP^Sc^ extraction from lymphoid tissue [127]. Although initial detection studies were performed with rodent models of scrapie [110,112,115,116,125,128,129] or with scrapie-affected sheep [124,126,130,131,132,133] distinct prion strains in different species have been found in lymphoid tissue, with it being necessary in some cases to use PrP^Sc^ concentration procedures or in vitro prion propagation techniques to enhance sensitivity. Transmissible Mink encephalopathy, which gives rise to two strikingly different strains upon passage in hamsters, is one of the clearest examples that some prion strains may show lymphotropism (i.e., Hyper (HY)) while others are more limited to the CNS (i.e., Drowsy (DY)) [123,134]. The original prion in mink shows early affectation of lymphoid organs by intracerebral inoculation. Upon oral inoculation, PrP^Sc^ is detected even earlier in retropharyngeal and mesenteric lymph nodes, then in spleen and finally in GALT. Moreover, in contrast to intracerebrally inoculated animals, PrP^Sc^ was found in rectal mucosa-associated lymphoid tissue (RAMALT) for orally inoculated minks [51]. Spleens, several lymph nodes, and myenteric plexuses from cats that developed Feline Spongiform Encephalopathy (FSE) after ingestion of BSE-contaminated foodstuff, were analyzed by IHC. PrP^Sc^ was only detected in few of these samples (2 out of 13 spleens, 1 out of 2 Peyer’s patches, 4 out of 4 myenteric plexuses and in all kidneys examined) [135]. In part, such inconsistent detection of PrP^Sc^ could be due to the low sensitivity of direct detection methods and the result might change if cell-free propagation systems were used. In fact, a study performed in a captive cheetah affected by FSE using a specific PrP^Sc^ precipitation protocol and IHC or immunoblot, showed prions in spleen and mandibular lymph nodes, confirmed also by PMCA propagation of spleen prions [136]. The presence of BSE prions in lymphoid tissue of different species has been also investigated due to its socioeconomic importance. Although not as clearly characterized as in the case of scrapie, in naturally occurring BSE cases, involvement of the lymphoreticular system appears to be low and infectivity has only been demonstrated in Peyer’s patches of the ileum by ultrasensitive bioassay. In experimentally infected cattle with high oral dose also could be detected in Peyer’s patches and inconsistently in tonsil [137]. However, the lack of detection could be due to the low sensitivity of direct detection methods and bioassays, since a study based on PMCA propagation showed PrP^Sc^ in palatine tonsils, lymph nodes, ileocecal region and spleen of cattle experimentally infected by the oral route [138]. The distribution of BSE in sheep has been also studied by IHC in pre-clinical animals, showing similar pattern to that of scrapie in sheep. Upon oral inoculation, PrP^Sc^ could be first detected in tonsil and ileal Peyer’s patches as early as six months after oral inoculation. At nine months, all the GALT and lymph nodes, including spleen, showed PrP^Sc^ and it was also detected in the peripheral nervous system. It appeared first in the enteric nervous system and from there invaded the CNS through the dorsal motor nucleus of the nervus vagus. At 12 months, and up to the development of clinical signs at 20–24 months post-inoculation, PrP^Sc^ spread within the lymphoid system to all non-gut-associated lymphoid tissues and completely invaded the CNS via the enteric nervous system through the parasympathetic and sympathetic nerves to the medulla oblongata and the spinal cord [139,140]. Similar studies performed by bioassay in sheep bearing other polymorphic PrP showed a generally similar pathway although with slight differences; for example the early involvement of retropharyngeal lymph nodes instead of tonsils [141,142]. However, in many cases detection depended on the thorough screening of several tissue sections by IHC, due to the low amounts of PrP^Sc^ present at early stages in lymphoid or enteric tissue. Thus, it is likely that the observed variability is derived from the different and low sensitivity of the detection methods used. Experimental infection of BSE and sheep-BSE in pigs has been also investigated using IHC and PMCA to detect minute amounts of prions in peripheral tissues. In this experiment, pigs were infected intracerebrally. In spite of the route of inoculation, prions could be detected already in the clinical stage of disease in palatine tonsils in some cases and in few submandibular and mesenteric lymph nodes but rarely in spleens or GALT, suggesting a centrifugal spread of intracerebrally inoculated prions from the CNS to the periphery, with low affectation of lymphoid organs [143,144]. Finally, another animal prion disease in which PrP^Sc^ detection in lymphoid tissues is relevant for diagnostic purposes is CWD of cervids, known for the high rate of horizontal transmission likely due to abundant presence of prions in extraneural tissue and body fluids [145]. PrP^Sc^ distribution in orally CWD-infected deer checked by IHC reveals its presence at 42 days post-infection in lymphoid tissues draining the oral and intestinal mucosa (retropharyngeal lymph nodes, tonsils, ileal Peyer’s patches and ileocecal lymph nodes), but not in other lymphoid organs, such as the spleen, mesenteric nodes, and others at 80 days post-infection [146]. Other studies using cervid samples post-mortem, or in more advanced stages of the disease, and in some cases using more sensitive techniques, showed PrP^Sc^ presence in other lymphoid organs, such as the RAMALT, demonstrating that the presence of prions in lymphoid tissues follows a spatiotemporal distribution similar to that of scrapie in sheep [48,147,148,149]. Apart from the detection by IHC or immunoblot, the necessity of limiting the spreading of naturally occurring prion diseases such as BSE, scrapie and CWD, resulted in the development of rapid diagnostic tests. First for BSE and afterwards adapted to small ruminants, they are mainly used for active surveillance by government agencies due to their shortened assay times, although with no significant improvement over standard methods on specificity and sensitivity (e.g., TeSeE^TM^ sheep/goats ELISA, Bio-Rad, Hercules, CA, USA; Prionics^TM^-Check PrioStrip SR, Prionics, Zurich, Switzerland). These tests, based on detection of PrP^Sc^ by ELISA were initially approved just for post-mortem analysis of brain samples [150], but the numerous reports on ante-mortem detection of prions in lymphoreticular system of sheep and cervids [131,151,152,153,154,155] prompted their use on lymphoid tissue as well as on brain samples [156]. Lymphoid tissues of choice, due to accessibility and ease of sampling and their association with the gastrointestinal track, are retropharyngeal lymph nodes and tonsils [151,157,158]. However, in sheep [159,160,161] and mainly in deer, another lymphoid tissue or its associated mucosa has been also widely used, namely the RAMALT [149,162]. For this tissue, cell-free prion propagation systems are being adapted in order to increase sensitivity and specificity [163,164,165,166,167]. Overall, the use of lymphoid tissues for the ante-mortem diagnosis of orally transmissible animal prion diseases is widely used together with brain samples for surveillance by government agencies.

### 4.2. PrP^Sc^ in Lymphoreticular System in Human Prion Diseases

The involvement of the lymphoreticular system for some human prion diseases was reported early on in the course of transmissibility studies to primates. Infectivity in spleen and lymph nodes of some patients with prion disease was demonstrated although irregularly, with those affected by Kuru showing the best transmission rates [80]. The discovery of variant CJD, the third acquired prion disease in humans derived from dietary exposure to BSE prions, prompted the search for accurate ante-mortem diagnostic methods in order to control its spreading. For that, based on observations with animal prion diseases and the fact that infectivity was found in lymphoid tissues in human prion disease, the presence of PrP^Sc^ in tonsillar biopsies of vCJD-affected patients was first explored [168]. In samples of deceased patients, PrP^Sc^ was found on tonsils by IHC and immunoblotting, suggesting that biopsy of lymphoid tissue could be a good method for early diagnosis of vCJD. In a comparative study between vCJD and other human prion disorders, PrP^Sc^ was detected in tonsils, spleens and lymph nodes of vCJD-affected patients but not in iCJD, indicating that tonsillar biopsy could be useful at least in advanced stages of disease [169]. The first notion that detection of prions in lymphoid tissue could be done ante-mortem came from the analysis of an appendix of a vCJD-affected patient, which was removed approximately one year before the onset of the clinical signs and where prions were detected by IHC [170]. By bioassay in mice, presence of infectivity in spleen and tonsil were also confirmed [171]. With an improved method for PrP^Sc^ detection based on selective precipitation and immunoblotting, prion distribution in four vCJD patients was analyzed finding that tonsils, spleens, and lymph nodes were uniformly positive for PrP^Sc^ (concentrations of 0.1 to 15% of those in brain), with tonsils showing consistently highest concentrations. In one of them very low levels of PrP^Sc^ were found in rectum, adrenal gland, and thymus, but all were negative in the appendix [172]. A similar study using another precipitation method showed also detection of vCJD prions in tonsils, in accordance with the previous one [173]. Although vCJD was considered to be unique in the involvement of the lymphoreticular system, in contrast to sporadic or genetic forms of the disease, the enhanced precipitation-based detection method was applied to the study of tissue distribution of PrP^Sc^ in sporadic CJD. This study revealed prions in 10 out of 28 spleens of confirmed sCJD patients, which were those showing the longest disease duration and the more uncommon molecular subtypes, correlating PrP^Sc^ presence in extraneural tissues with long disease duration [174]. Technical improvements of the analytical methods offering enhanced sensitivity for PrP^Sc^ detection, however, showed complete absence of prions in lymphoid tissues of sCJD cases in contrast to vCJD, where PrP^Sc^ was found in spleens, tonsils and lymph nodes [175]. In order to shed some light on the presence of peripheral PrP^Sc^ in the distinct human prion diseases, primate models were inoculated intracerebrally and via intratonsillar route with different tissue homogenates from vCJD, sCJD, and iCJD patients. In this study, prions were detected in spleen, tonsils, mesenteric lymph nodes, and Peyer’s patches from the small intestine of vCJD and BSE inoculated animals, while for sCJD, prions were detected in low amounts in the spleen, and for iCJD in tonsils, but not in other lymphoid tissues. These results definitively confirmed vCJD as a highly lymphotropic strain whereas other CJD types show very low lymphoid tropism [16]. A similar study using primate models, orally inoculated with BSE prions, also showed widespread presence of PrP^Sc^ in lymphoreticular system (tonsils, spleen and ileocecal lymph nodes) at the clinical stage. However, prions were rarely detected in these organs in preclinical stages, indicating caution for assessment of vCJD prevalence in the population based on screening of lymphoid tissues which could greatly underestimate the number of affected individuals [176]. Nonetheless, a large scale prevalence study performed on more than 30,000 appendices of UK citizens by IHC revealed PrP^Sc^ in one out of 2000 persons, pointing towards a much higher prevalence than expected and with important implications for public health [177]. Another one conducted on more than 10,000 tonsil specimens from the UK only detected PrP^Sc^ in one sample, offering prevalence data for vCJD strikingly different of that predicted in the study done with appendices [158]. In some cases, humanized transgenic mice have been employed to determine the PrP^Sc^ containing tissues and evaluate transmission risk. For example, lymphotropism of atypical BSE was evaluated through inoculation in mice expressing human PrP, in which PrP^Sc^ was detected in spleen [178]. Another study aimed to determine if the PrP^Sc^ detected in spleen and lymph nodes of a recipient of vCJD-infected blood that never developed disease (likely due to being heterozygous for codon 129), was infectious and thus a supposed risk of transmission. Since the mice succumbed to infection when inoculated with spleen tissue of the subclinically affected patient, they demonstrated that the PrP^Sc^ present in the spleen (without CNS involvement) was also infectious and able to cause disease in susceptible recipients [179]. Finally, studies in transgenic mice also allowed the quantification of prion infectivity in different tissues by endpoint titration experiments. Infectious titers calculated in spleen of a vCJD-affected patient using this method were of ~10^6.1^ ID_50_/g, although its use for diagnosis is highly limited due to the duration of the assay [180].

Given that in humans, sporadic and genetic prion diseases are far more common than acquired prion diseases and that their lymphoreticular involvement is generally low or absent, few efforts have been made for the adaptation of highly sensitive PrP^Sc^ detection methods, such as PMCA or RT-QuIC to these tissue samples. However, the use of rapid screening tests based on ELISA that are commonly used for animal TSE surveillance, has been explored with lymphoid tissues (in addition to brain homogenates) for human prion disease associated PrP^Sc^ detection. Tonsils and spleens of few vCJD-affected patients were analyzed and found to be positive for prions in all cases, demonstrating the potential of such rapid tests for vCJD diagnosis [181]. Finally, cell-free prion propagation systems have been used to further assess PrP^Sc^ distribution in peripheral tissues of sCJD patients in comparison with vCJD patients, since the lack of infectivity in the peripheral organs of sCJD-affected patients could be due to an insufficient sensitivity of previously applied techniques. In contrast to many previous reports claiming absence or an irregular detection of PrP^Sc^ in lymphoreticular system of sCJD patients, the ultrasensitive method used in this study detected similar levels of PrP^Sc^ in spleens, lymph nodes and tonsils of sCJD and vCJD affected patients [182].

In summary, lymphoid tissues and associated mucosa are widely used for the diagnosis of animal prion diseases such as scrapie and CWD. Since rapid tests based on ELISA offer good specificity and sensitivity for these cases, cell-free prion propagation techniques have not been extensively used, despite showing enhanced sensitivity compared to the previous methods. However, in the case of human prion diseases, analysis of lymphoid tissue could be useful for vCJD detection but has not been exploited for sporadic or genetic TSE due to the putatively low and late affectation of lymphoreticular system. Moreover, even in the case of vCJD in which patients with extremely low involvement of lymphoid organs have been reported [183], the use of these tissues for early diagnosis could be challenging, favoring other strategies as detection in CSF or other more accessible body fluids.

## 5. Detection of PrP^Sc^ in Blood

The presence of prions in blood and specially infectivity in this body fluid has been a cause of concern since the transmissibility of prion diseases was established. In fact, in experimental rodent models of scrapie, infectivity in serum was shown as early as 1960s [184,185]. However, only few of the initial transmission experiments in animal models confirmed such results, being positive only for some blood fractions using certain rodent models of scrapie [186,187,188]. Similar results were found for human CJD and GSS blood samples inoculated by different routes in rodent models [189,190,191], while many other similar studies showed negative results for blood infectivity [192,193,194,195,196]. Despite the variable results, the possibility of TSE transmission through blood had been established experimentally at least for some prion diseases and the scarce evidences that in human prion diseases could also occur [197], set the alarms ringing for the risk of transmission via blood transfusion [198]. The outbreak of vCJD in the UK further increased the urgency to definitively assess the infectious potential of blood and blood-derived products from pre-symptomatic TSE-affected individuals, to develop sensitive methods for prion detection in blood and procedures that could reduce the risk of contamination during the processing of blood-derived products [199,200,201,202]. At the same time, researchers started to explore the extraneural distribution of prions due to the discovery of the involvement of the lymphoreticular system and its close association with the circulatory system. The blood fractions containing prion infectivity were mainly sought given the importance for the control of possible prion contamination in blood-derived products [191,202,203,204,205,206]. From these studies, it could be deduced that infectivity in blood is much lower than that found in brain (10^5^-fold lower), that it is mainly present in leukocytes (5 to 10-fold more than in plasma or red cells) and that for some prion disease infectivity in blood could be present as early as half-way the incubation period [207].

### 5.1. PrP^Sc^ in Blood According to Transmission Studies

Based on those observations and suspecting that lack of infectivity detection in many experimental transmission experiments was due to the low sensitivity of the available techniques, several sophisticated methods for PrP^Sc^ detection in blood were developed using animal models of prion infection or blood spiked with brain material from TSE-affected individuals [208,209,210,211,212,213,214,215,216,217,218]. The report of the first cases of vCJD transmission through contaminated blood transfusion in 2004 [219,220,221] entailed a point of inflexion, since a theoretical hazard became real, causing a significant increase in researchers dedicated to investigate infectivity and PrP^Sc^ presence in blood as well as improved detection methods. In all cases, blood transfusion recipients developed vCJD after receiving transfusions of red blood cells from donors that were pre-symptomatic at the time of donation but developed vCJD later. Transmission studies in animal models continued and further complicated the situation, using different blood fractions to assess infectivity levels. These studies confirmed the presence of infectivity in whole blood and buffy coat of scrapie and BSE-affected sheep [222], in BSE-inoculated mice, and primate models [223,224] as well as in GSS and vCJD-inoculated mouse models [225], among others [198]. In addition, for CWD in cervids, known for its ease of natural transmission due to the presence of prions in most of the tissues and body fluids analyzed to date, infectivity in blood was definitively proven by the intravenous (equivalent to transfusion) route in symptomatic [226] and pre-symptomatic animals [227]. The infectivity was later described to be harbored by B lymphocytes and platelets [228] as found also in scrapie-infected sheep [229]. Nonetheless, other studies with mouse models and vCJD prions showed that all clinically relevant blood fractions can contain prions [230], as well as with naturally acquired scrapie in sheep [231] and that the efficiency of blood to blood transmission could be higher than expected [232]. Although the situation with vCJD was increasingly clear (transmission risk had been established and the dependence on the recipient’s genotype regarding codon 129 for the development of the disease had been observed), the risk of transmission of other human TSE types, such as sporadic or genetic cases was still under study [207]. Despite some evidences of the presence of PrP^Sc^ in blood or plasma from sCJD patients and animal models infected with sCJD or GSS prions [216,233,234], there are also reports of lack of infectivity or PrP^Sc^ [182,234,235]. On the other hand, no cases of infection through blood transfusion have been reported in humans and epidemiological and follow-up studies suggest that the possibility of occurrence of such an event is extremely low [236,237,238,239]. However, a recent report of neurological impairment after vCJD prion contaminated-blood infection in mice and macaques suggests, that apart from a low percentage of vCJD transmission, another neurological condition could be transmitted through transfusion of contaminated blood [240], reinforcing the importance of developing detection methods for prions in blood. In addition, experimental inoculation of mouse-adapted CWD prions by the intravenous route gave rise to a novel more neuroinvasive strain, that indicates that exposure to prions through transfusion could generate new prion strains in humans [241].

### 5.2. PrP^Sc^ in Blood According to New Detection Techniques

Despite the development of new detection methods with increased sensitivity, in many cases resulted insufficient and reports of lack of PrP^Sc^ detection in blood fractions that have been proven positive in infectivity studies were still published [172,242]. A major breakthrough in this matter occurred in 2005 when cell-free prion propagation methods were applied to the detection of prions in blood. PMCA was shown to detect prions in buffy coats from scrapie-infected hamsters with unprecedented 89% sensitivity and 100% specificity [21]. Moreover, this methodology was readily improved and used for the detection of minute amounts of PrP^Sc^ in blood of pre-symptomatic animals, detecting it in scrapie-infected hamsters down to 20–50 molecules of misfolded PrP, with a maximum sensitivity of 60% and specificity of 100% in the pre-clinical phase (at 40 days post-inoculation) and reaching 80% sensitivity in the clinical stage (at 115 days post-inoculation) [243]. These results brought some hope to the possibility of detecting prions for the screening of blood for transfusion or other blood-derived products and importantly, for early diagnosis of prion diseases using an easily accessible body fluid. In fact, apart from the development of new PrP^Sc^ detection methods for diagnosis, PMCA has permitted to detect PrP^Sc^ in extracellular vesicles from plasma of vCJD infected mice [244], which was later confirmed by IHC [245]. Although other methods have been proposed (e.g., rapid tests based on ELISA [18,246], cellular assays using prion susceptible cells [247], or bioassay in Drosophila [248]), the worse specificity and sensitivity and difficulties associated to cell culture or fly models for standardization purposes have limited their use. Nonetheless, several laboratories and companies have developed their own methods for screening of vCJD prions in blood based on distinct procedures. This is the case of the EP-vCJD blood screening assay from Amorfix Life Sciences Ltd., based on prion concentration followed by immunoassay, which showed 100% sensitivity and 97%–99% specificity with brain prions spiked in blood and plasma [249]. Other systems have been also developed not only based on in vitro prion propagation, with variable sensitivity and specificity aimed to the detection of vCJD or sCJD prions in blood. These methods have been tested in different models (summarized in Table 2), but in general, procedures based on cell-free prion propagation such as RT-QuIC and PMCA can be considered the most promising due to their versatility. They have been applied to blood in several occasions and with different prion strains and species, offering maximum specificity and overall high specificity (Table 2). Although there is no routine blood test fully developed yet, blood analysis could be one of the easiest way for the early diagnosis of prion diseases as long as sufficient sensitivity is achieved and the ability to detect not only vCJD, but other human prion diseases at the pre-clinical stage is definitively demonstrated. This might not be far off since the use of recombinant bank vole PrP as RT-QuIC substrate has shown the capacity to propagate all human prions tested from brain homogenates [42]. Meanwhile, analysis of surrogate biomarkers in blood is becoming a feasible strategy to aid in early diagnosis of TSE, given that total PrP and total tau levels have been already correlated with CJD [250,251].

## 6. Detection of PrP^Sc^ in Olfactory Mucosa

In the previous sections it has been clearly established that despite being pathologies of the CNS, prions can replicate in extraneural tissues (such as lymphoid organs) and that they can spread to different locations through the peripheral nervous system [118,266]. Moreover, intraocular inoculations done to study the neural spread of prions [267] and the report of iatrogenic CJD cases due corneal transplants [268] clearly demonstrated that prions could spread in both directions along the axons. Further demonstration of PrP^Sc^ spreading along the sensory pathways and cranial nerves came from inoculation studies using the intratongue route [123] and from PrP^Sc^ deposition assessments that showed detectable amounts of prions by IHC in the optic nerve of a sCJD-affected patient [172]. The possibility of finding PrP^Sc^ in peripheral nerves that could be accessed easily and that were close to the CNS, lead Zanusso and collaborators to pay attention to the olfactory sensory pathway, which is in close contact to the CNS through the olfactory receptor neurons. The dendrites of these neurons are located in the olfactory epithelium of the upper nasal cavity form the olfactory cilia, which is directly connected to the olfactory bulb. In fact, they were able to determine in post-mortem sCJD samples, that prions were selectively deposited in the neuroepithelium of the olfactory mucosa, which could be detected by IHC and WB analysis [269]. Although early involvement of the olfactory system was suspected, indicating it could be useful for early diagnosis of sCJD, it was not proven until a year later. The same group was able to detect PrP^Sc^ in a patient just 45 days after disease onset although just by IHC likely due to low levels for immunoblot detection [270]. The presence of prions in olfactory mucosa in early stages of sCJD that could be due to selective centrifugal spread of prions from the brain via olfactory sensory pathway, opened the way for a non-invasive early diagnosis of TSE based on PrP^Sc^ detection in nasal brushing.

In animals, the shedding of prions to oral and nasal mucosa was investigated by laser scanning confocal microscopy in hamsters inoculated intracerebrally with the HY prion strain. They observed PrP^Sc^ in the olfactory and vomeronasal epithelium, mainly in the apical dendrites of the sensory neurons in early clinical stage of disease, confirming transport of prions via the olfactory nerve fibers that descend from the olfactory bulb early in the disease [271]. The involvement of the olfactory system has also been investigated in sheep naturally affected by scrapie. In this case, PrP^Sc^ was detected by IHC or immunoblotting in nasal cavity and olfactory system-related brain areas. They were able to detect PrP^Sc^ in the nasal cavity of 21 out of 24 sheep samples analyzed in both pre-clinical and clinical stages of disease. However, olfactory receptor neurons and olfactory epithelium, analyzed just by IHC, did not show PrP^Sc^ deposition, likely due to the small amounts present (as in humans in which was estimated to be around 3% of that in the olfactory bulb) or the high turnover rate of olfactory receptor neurons [272].

### Cell-Free Prion Propagation Systems and PrP^Sc^ Detection in Olfactory Mucosa

The low PrP^Sc^ levels in olfactory mucosa could have greatly limited its use for diagnosis if cell-free prion propagation techniques had not come onto the scene. In fact, further studies in which prion shedding to the olfactory mucosa of hamsters was reported, already took advantage of the RT-QuIC system to confirm the previous findings, being the first reports on the use of nasal brushings for prion detection [273,274]. The use of RT-QuIC with olfactory mucosa obtained from nasal brushings was readily applied to the detection of human prions in a study that compared the results of the olfactory mucosa with samples of CSF from the same patients. Tested in definite and probable sCJD cases, as well as in a couple of gCJD cases, the assay showed 97% sensitivity for sCJD and 75% for gCJD and 100% specificity, improving the sensitivity offered by CSF analysis of the same patients that was around 77%. Moreover, nasal brushings provided stronger and faster seeding than CSF samples in RT-QuIC, since they contained significantly higher prion concentrations than CSF samples [96]. Therefore, nasal brushing RT-QuIC could be more promising for TSE diagnosis than the CSF-based technique, although how long before disease onset prion shedding to the olfactory mucosa occurs needs to be addressed for that.

This technique has been also applied to the analysis of CWD-affected deer and elk samples in comparison with RAMALT tissue obtained ante-mortem from clinically affected animals. In this case, while RAMALT sensitivity was around 70% and specificity of around 94%, nasal brushings performed much worse, with sensitivity reaching 15% and specificity ranging from 90% to 100% in white-tailed deer [165]. In ante-mortem and post-mortem samples of elk, similar results were obtained with RAMALT reaching 78% sensitivity, in contrast to the 34% achieved with nasal brushing [166]. Few other studies have been performed in humans due to the novelty of the method, but the published studies indicate that nasal brushings coupled to in vitro prion propagation techniques could be one of the most promising diagnostic tests in development that use PrP^Sc^ as pathognomonic biomarker. Both PMCA and RT-QuIC on nasal brushings of two FFI patients demonstrated that in this genetic prion disease, PrP^Sc^ can also be found in trace amounts in olfactory mucosa samples. Both techniques were able to detect as little as femtogram amounts of PrP^Sc^ in nasal brushings of FFI-affected patients, with 100% specificity and quantitative PMCA allowed to estimate PrP^Sc^ concentration on these samples of 1.41 × 10^−14^ g/mL [275]. Another control-case study in olfactory mucosa was performed with samples from 86 CJD patients (probable, possible and suspected cases) versus 104 controls analyzed by RT-QuIC. The results showed that from 61 patients with definite sCJD, 93% to 100% sensitivity was achieved with 100% specificity, whereas for the 8 gCJD or GSS cases, 75% sensitivity was reached with 100% specificity and suggested that the combined CSF and olfactory mucosa analysis could provide virtually 100% sensitivity [102]. In the most recent study, 182 CSF samples and 42 olfactory mucosa brushings from patients suspected of having sCJD with rapidly progressive dementia were submitted to RT-QuIC showing sensitivity of 91% (versus the 96% achieved with CSF samples) and 100% specificity, significantly higher than surrogate biomarkers 14–3-3 and total tau for the same samples [105]. Again, the study concludes that combined CSF and olfactory mucosa analysis by RT-QuIC can achieve 100% sensitivity in sCJD, proposing it as one of the possible ways to develop the definitive test for TSE diagnosis in humans. Finally, it is worth mentioning that the success of this method for the diagnosis of TSE has prompted its use on other protein misfolding-related neurodegenerative disorders such as Parkinson’s disease or Multiple System Atrophy [276].

## 7. Detection of PrP^Sc^ in Urine, Saliva, and Feces

Prion diseases with high rate of horizontal transmission such as scrapie and CWD are well-known for the spreading of PrP^Sc^ in a wide range of peripheral tissues and body fluids. One of the possible explanations for their considerable dissemination rates was long suspected to be due to the excretion of prions through body fluids that persist in the environment and are afterwards acquired by other individuals. In fact, in contrast to other orally acquired prion diseases such as BSE, scrapie-affected animals show PrP^Sc^ throughout their digestive track including colon [277], indicating that excretion in feces could be a route for horizontal transmission. However, although the amounts of PrP^Sc^ might be much higher in scrapie or CWD it does not seem an exclusive trait of these prion diseases as demonstrated by the detection of protease-resistant PrP forms in urine of BSE-affected cattle, sCJD or gCJD-affected patients as well as in intracerebrally scrapie-infected hamsters in pre-clinical stages of disease [278]. Based on the observations that prions could be found in urine, probably in feces and likely in other body excretions, the search for a non-invasive diagnostic test based on prion detection in these fluids started.

### 7.1. Urine

The initial attempts to detect prions in urine of CJD affected patients, aimed to developing a diagnostic test, failed to detect proteinase K-resistant PrP by immunoblotting [278] due to the presence of several protease-resistant proteins coming from contaminant Enterobacterial species [279]. A second study using urine samples from sCJD, gCJD, and vCJD-affected patients also reported detection of multiple protese-resistant proteins in the range expected for prions that showed cross-reactivity with anti-PrP antibodies but consisted fundamentally of immunoglobulins from the urine. Despite partially solving this problem by using two distinct secondary antibodies, the assay showed overall low sensitivity and specificity: 31% with one of the antibodies and 64% sensitivity with the other one for sCJD, 0% and 33% for vCJD, and 25% and 50% for the gCJD cases, with specificity of 88% and 35%, respectively [280]. Even a third one was reported in which the same PrP-immunoglobulin composed aggregates were observed in concentrated urine samples of CJD-affected patients, failing in this occasion to determine whether urine could be used for human TSE diagnosis [281]. In spite of the low performance of these initial assays the presence of PrP^Sc^ in urine was established and thus, studies aimed at using PrP^Sc^ detection for diagnosis or for transmission risk assessment increased [282]. Prionuria or presence of prions in urine was found to be increased due to nephritis in scrapie-inoculated mouse models. By bioassay they detected infectivity in urine of mice affected by this condition but not in prion-infected mice not suffering the renal pathology [283], that could reflect the low titers present in animals not suffering from nephritis. In agreement with this observation, a couple of studies performed with scrapie-infected hamster urine showed that inoculation with urine caused subclinical infection in most cases, developing into scrapie just in few animals [284,285]. Another bioassay in humanized transgenic mice inoculated with raw and 100-fold concentrated urine from three sCJD-affected patients also showed negative results, indicating that there was not PrP^Sc^ in the urine of these patients or that the amount was lower than 0.38 infectious units/mL (based on titration experiments with brain-derived sCJD prions in the same animal model) [286].

Again, specific procedures designed ad hoc and in vitro prion propagation systems contributed to solve the problems associated with the low PrP^Sc^ amounts present in urine samples, confirming presence of PrP^Sc^ in urine of several species and opening the way for the development of diagnostic tests based in urine. For instance, a method developed originally for detection of PrP^Sc^ in blood of vCJD patients named direct detection assay and based on PrP^Sc^ concentration [264], was adapted to urine samples. Using this procedure, the authors reported 40% sensitivity on detecting PrP^Sc^ in urine of sCJD patients but only around 8% for vCJD samples of patients within the symptomatic stage of the disease, both with 100% specificity [287]. However, cell-free prion propagation methods have provided the best results taking into account the low concentrations of PrP^Sc^ in urine. PMCA has been used to detect prions in urine of orally scrapie-infected hamsters, revealing that in few animals PrP^Sc^ was excreted in urine four days after inoculation, although afterwards it was undetectable until the final stages of disease [288]. An initial estimation of sensitivity and specificity of PMCA for the detection of prions in urine of scrapie-infected hamsters in the clinical phase of disease showed approximately 80% sensitivity and 100% specificity, although larger number of samples might be required for an accurate calculation of sensitivity [289]. Variations of the PMCA methodology in order to enhance sensitivity have been used for the specific detection of atypical BSE prions in urine of experimentally infected macaques. Using L-Arginine ethylester during the in vitro propagation, the technique allowed detection of prions in urine of 1 out of 2 animals at pre-clinical stage and in all the samples at the clinical stage of disease [85]. PMCA has also been applied to the study of scrapie-affected sheep and CWD-affected deer urine samples. Rubenstein and colleagues reported PrP^Sc^ detection in urine of sheep naturally or experimentally infected with scrapie at the clinical stage of disease and for deer infected with CWD in pre-clinical, early and late clinical stages of the disease consistently by PMCA [290]. Similarly, another study on CWD-affected cervids showed detection of prions in urine by PMCA and bioassay although at lower levels than in saliva [23]. Importantly, it has also been applied to determinate the time course of prion excretion in urine of three different orally CWD infected-cervid species at the preclinical stage: elk, mule deer, and white-tailed deer. They found that prion excretion started as early as 6 months post-infection in feces whereas their detection in urine resulted 10 times less frequent, finding PrP^Sc^ at six months only in one out of two white-tailed deer and requiring 18 months to be detected in one of the two elks and in the only mule deer tested [291]. The use of PMCA for the detection of prions in urine of human TSE-affected patients has also been explored. Moda and collaborators analyzed urine samples from several patients in the clinical phase of vCJD, sCJD, gCJD, or other neurological disorders, and found PrP^Sc^ only in vCJD patients’ urine with a sensitivity of 93% and a specificity of 100% [292]. Furthermore, they demonstrated that urine-derived PrP^Sc^ propagated by PMCA conserved its strain features by bioassay in humanized transgenic mice [293]. Regarding RT-QuIC, its utility on urine prion detection has also been proven, mainly applied to CWD samples. In white-tailed deer and mule deer orally inoculated with CWD-containing brain homogenate, urine from the pre-clinical stage was analyzed, being negative for PrP^Sc^ at five months post-infection but consistently positive at 13 and 16 months, three to four months before development of the first clinical signs [294]. Quantification of PrP^Sc^ amount was also performed by RT-QuIC and using a specific prion precipitation procedure to improve sensitivity. Results showed similar PrP^Sc^ amounts in urine and saliva of symptomatic deer of approximately 1–5 LD_50_/10 mL [295]. The same authors also presented a longitudinal study in CWD-infected white-tailed deer to determine the onset, duration, and amount of prion shedding in urine. They detected PrP^Sc^ in urine by RT-QuIC at 6 months post infection and estimated infectivity titers were found to be coincident with the previous study [296]. An enhanced RT-QuIC procedure for the analysis of urine samples has been also presented, based on PrP^Sc^ extraction and purification using magnetic beads. This method, also applied to CWD prions, showed similar sensitivity in urine than other methods involving precipitation procedures [295], but the one based on magnetic beads improved assay consistency (reduced range among replicates) [297]. Apart from CWD prions, RT-QuIC has also been used to quantify PrP^Sc^ in urine of mice inoculated with mouse-adapted scrapie prions. Importantly, this assay allowed the monitoring of the efficacy of an anti-prion therapy in these model animals, in which brain PrP^Sc^ amounts correlated with the amounts found in urine [298]. Therefore, in addition to diagnosis, cell-free amplification techniques may provide the perfect platform to monitor the effect of therapeutic agents in efficacy assays using a non-invasive method.

### 7.2. Saliva

The presence of prion infectivity in saliva was found first in CWD-affected cervid by bioassay, looking for explanations for the high rate of horizontal transmission that characterizes this TSE [226]. In scrapie-infected goats, PrP^Sc^ was also observed in salivary glands by immunoblotting in amounts ranging from 0.02% to 0.005% of the amount detected in the brain [299]. These observations were further confirmed in another bioassay study using TSE-affected deer and sheep saliva. In this case, six out of seven urine samples from scrapie-infected sheep transmitted prions to ovinized transgenic mice with titers of −0.5 to 1.7 log ID50 U/mL. Similarly, inoculation of saliva samples from two CWD-affected mule deer transmitted prions to mice expressing elk PrP with titers of −1.1 to −0.4 log ID50 U/mL [300].

Due to the low amounts, detection of PrP^Sc^ in saliva was definitively confirmed thanks again to cell-free in vitro prion propagation techniques. However, in the first attempts, results of CWD-prion presence in saliva determined by bioassay could not be further confirmed by PMCA, in contrast to presence in urine in which PrP^Sc^ was readily detected, most likely due to propagation inhibitors present in the saliva [23]. Inhibition problems were also observed when CWD-affected deer saliva was used as seed in RT-QuIC and mucins from saliva were identified as inhibitors. Given the observation that sonication could degrade part of the RT-QuIC inhibitors, Davenport and colleagues decided to try a modified PMCA protocol (increased temperature and energy input) that allowed detection of PrP^Sc^ in samples that were negative for RT-QuIC. Nonetheless, PMCA requiring multiple serial amplification cycles is time-consuming and the authors decided to combine both methods to reduce assay times. For that, they used PMCA products still undetectable by immunoblotting to seed RT-QuIC reactions [301]. PMCA has also been exploited for the detection of BSE-infected cattle, although enhanced methods were also required in order to increase sensitivity. The use of sulfated dextrans to specifically favor propagation of BSE prions in vitro, allowed the detection of PrP^Sc^ in salivary glands and saliva of BSE-infected cattle at terminal stage of disease with 50% sensitivity (two out of four samples were positive) [138]. The same method was applied to the detection of PrP^Sc^ in saliva of cattle at terminal stage, early clinical stage and two months before onset of clinical signs. The authors estimated salivary PrP^Sc^ amount to be equivalent to that in 10^−12^ dilution of brain homogenate, however failed to detect in cattle three to five months prior to the onset of the disease [302]. Another modified PMCA protocol using L-Arginine ethylester also allowed early detection of prions in saliva as well as in urine of atypical BSE-infected macaques. PrP^Sc^ presence was shown in saliva in half of the samples obtained from one animal at pre-clinical stage of the disease and in all the samples of another animal at the symptomatic phase of the disease [85]. These studies clearly indicate the importance of optimizing each assay for specific prion strains and PrP^Sc^ seed source. RT-QuIC, once inhibition problems were solved through dilution of saliva samples or using prion-enriched samples precipitated with phosphotungstic acid, also allowed detection of PrP^Sc^ in saliva obtained from CWD-exposed white-tailed deer. Specifically, prions were detected in 14 out of 24 (58.3%) diluted saliva samples, including 9 out of 14 asymptomatic animals (64.2%) and in 19 out of 24 (79.1%) enriched samples [303]. The same assays were used to quantify prions in saliva samples of CWD-infected deer, detecting similar levels than in urine of 1–5 LD_50_ per 10 mL [295]. Moreover, for a longitudinal study in which prions were detected in saliva of CWD-affected deer as early as 3 months post inoculation with an assay specificity of 97% (a bit lower than in urine with 99% specificity) [296]. Further modifications of RT-QuIC assay, such as the PrP^Sc^ extraction with magnetic beads before the in vitro seeding reaction, showed sensitivity and specificity similar to that obtained by phosphotungstic acid precipitation in the previous study [297].

### 7.3. Feces

Presence of prions in fecal material was first suspected due to the detection of PrP^Sc^ in the gastrointestinal track by IHC [277]. Moreover, it has been repeatedly established that prions are not degraded by gut microbiota and do not lose infectivity after passage through the digestive system of different species [304,305,306,307]. The first experimental evidence of prion transmission through feces was reported in hamsters that were exposed to the bedding of orally scrapie-infected hamsters, which developed infection with 80–100% attack rate, likely due to coprophagy. Presence of prions in feces was confirmed by immunoassay and the amount determined by bioassay, showing that fecal samples collected from infected hamsters in the first seven days after oral challenge harbored 60 ng of PrP^Sc^ per g of feces and prion titers of approximately 10^6.6^ ID_50_/g [308]. Using a procedure involving detergent-based extraction and immunoprecipitation, PrP^Sc^ was also detected in feces from orally inoculated mice over four-days post-inoculation shortly after oral ingestion of scrapie and BSE agents (up to 24 and 48 h post-infection, respectively), although it failed to detect prions in feces from terminally sick scrapie-infected mice [309].

Since lack of prion detection at terminal stages of disease could be due to low sensitivity of the techniques used, cell-free prion propagation methodologies were readily applied to the analysis of fecal samples. Krüger and collaborators were able to detect PrP^Sc^ in feces of orally scrapie-infected hamsters by immunoblotting at 24–72 h post infection, but not at the first 24 h or at later preclinical or clinical stages. To detect some PrP^Sc^ in clinically affected animals PMCA was required, finding prions in two out of four animal feces tested [50]. PMCA and two different PrP^Sc^ extraction methods were also used to detect prions in feces of sheep naturally infected with scrapie, showing positive results in 7 of 15 sheep at clinical stage and in 14 out of 14 sheep at preclinical stage [310]. For the detection of CWD prions, an adapted PMCA was also presented able to detect 1.2·10^−7^ dilution of brain-derived CWD prions in fecal environment, equivalent to 100 pg of PrP^Sc^ per g of feces. This system was used to detect prions in feces of naturally infected elk and estimate the concentration in correlation of IHC findings in the obex and RAMALT of the same animals, observing a good relationship between the amounts detected in feces by PMCA and the stage of the disease as indicated by IHC [311]. Another study with three CWD-affected cervid species (elk, white-tailed deer, and mule deer) in which prions in feces were sought by PMCA showed positive results in 10 out of 12 elk samples, in seven of nine white-tailed deer, and in seven out of eight mule deer; detection being possible for all three species in some samples already at six months post-inoculation [291]. In contrast, RT-QuIC has been only used in fecal samples of CWD-affected animals. The enhanced RT-QuIC detection method based on magnetic particle extraction of prions that was also used to detect prions in saliva and urine was applied to feces from terminal CWD-affected deer. Five out of six (83%) samples from CWD-infected deer were positive as well as 3 out of 46 uninfected control samples (false positive rate of 6.25%). Whereas in crude samples for which PrP^Sc^ was not extracted using magnetic beads only one out of six (16.6%) CWD-infected samples were positive [297]. This result reflects the difficulties of performing in vitro prion propagation assays in complex substrate as fecal material, suggesting that previous purification or extraction steps can be of utmost important for the success of a detection method. Another method based on prion precipitation with phosphotungstic acid has also been developed for the screening of CWD infection in pre-symptomatic animals in RT-QuIC using fecal material. Using this method in experimentally CWD-inoculated elk from which feces were collected at different timepoints, out of the 14 preclinical samples obtained between 8 and 400 days post infection 11 samples (78.5% sensitivity) were positive by RT-QuIC, including at least one out of two samples taken at very early time points (8, 9 or 14 dpi) [312]. Further improvement of this method through substrate replacement during RT-QuIC increased the sensitivity (77%; 14 out of 18 samples in RT-QuIC were positive) and specificity (100%; none of the negative controls in RT-QuIC turned positive) of detection, demonstrating that there is still room for improvement for these kind of diagnostic tests [313]. Apart from using distinct extraction methods for PrP^Sc^ from fecal samples, the RT-QuIC parameters can also be modified to improve the assay sensitivity and specificity. Using magnetic bead-based prion extraction and low temperature RT-QuIC, Henderson and colleagues were able to reduce the false-positive reactions from 34.2% to 2.5%. With this modified reaction protocol, they were able to detect PrP^Sc^ in 81.8% of feces from deer experimentally infected with CWD for one year or longer and also in naturally affected elk in which 40% of asymptomatic but certainly infected animals could be identified [314]. Therefore, diagnosis of TSE through analysis of feces seems feasible at least for CWD, although the techniques used may need further optimization.

## 8. Detection of PrP^Sc^ in Other Tissues and Body Fluids

Due to the possibilities offered by cell-free prion propagation techniques, PrP^Sc^ has been detected in many other organs, tissues and body fluids. Although most of them are not adequate sources of prions for ante-mortem diagnosis of TSE due to their bad accessibility, late appearance of prions in the course of the disease or inconsistent detection, they are worth mentioning given that further improvements in prion propagation methods could allow their use for this purpose.

As mentioned before, the eyes have drawn the attention of TSE researchers due to the success of intraocular inoculations, reports of iCJD due to corneal transplant and retina being part of the CNS [267,268]. The presence of prions in retina has been reported in several species including humans [172,315,316]. In fact, a study performed using RT-QuIC in post-mortem sCJD patients’ samples, revealed PrP^Sc^ in cornea, lens, ocular fluid, retina, choroid, sclera, optic nerve, and extraocular muscle [317]. However, due to the invasiveness of a retinal biopsy it seems difficult to use it in routine diagnosis. Much more promising, at least for scrapie and CWD, is the nictitating membrane or third eyelid, closely associated to lymphoid tissue and thus, a site of prion replication. In fact, preclinical detection of prions in this tissue has been achieved by IHC [318,319,320] and RT-QuIC [321].

Given the generalized spread of prions throughout the body for certain prion strains and their presence in highly innervated tissues [266], skeletal muscle, and skin have been also tested for presence of prions. In skin, it seems to be associated to peripheral nerves rather than keratinocytes and has been detected by immunoblotting in scrapie-infected sheep [322], in patients with vCJD [323] and in sCJD patients by bioassay and RT-QuIC [324], and also by RT-QuIC in animal models infected with sCJD, even at pre-clinical stages [325]. Therefore, skin biopsy-based diagnostic test could be feasible soon. Regarding skeletal muscle, presence of prions has been detected in humans [174,326] in primate models of human prion diseases [16,327] and sheep [133] among others, although its use for diagnosis using in vitro prion propagation systems has not been considered.

The possibility of vertical transmission of TSE in natural conditions, which was confirmed by bioassay in BSE infected mice [328] took researchers to study the presence of PrP^Sc^ in placenta of infected animals early on. Infectivity in fetal membranes and placenta of scrapie-infected sheep has been demonstrated in sheep, goat and mice models by bioassay or immunological techniques [126,329,330,331], although transmissibility seems to be determined by the genotypes of the ewe and the lamb [332,333]. In this case, PMCA has been applied using placental tissue as source to determine the absence of PrP^Sc^ in case of fetuses showing scrapie-resistance alleles [334]. Similarly, infectivity in milk or semen has also been evaluated, despite initial attempts to detect prions in milk failed [328,335]. However, the presence of low levels of infectivity in milk of scrapie-infected sheep was finally demonstrated [336,337,338] and shown to be increased in sheep suffering mastitis [339]. Using PMCA, presence of PrP^Sc^ in milk of scrapie-exposed sheep was confirmed, even in the case of subclinical infections [340]. However, a diagnostic test based on placenta or milk would be only applicable to females limiting the development of such assays. Embryo transfer and artificial insemination are widely used in animal husbandry and, therefore, assessing presence of prions in embryos and semen could be of interest, although not for diagnostic purposes. Although the first bioassay study failed to transmit scrapie in to transgenic mice by inoculation of scrapie-affected ram semen [341], the possible natural infection of several ewes in a scrapie free-flock that had been bred with likely scrapie-infected rams led Rubenstein and colleagues to use PMCA to detect scrapie prions in semen of experimentally infected rams, demonstrating low levels of seeding activity [342]. The same methodology applied to semen samples of CWD-infected deer also confirmed the presence of prions in this body fluid, albeit with low sensitivity, probably due to the low titers [343].

## 9. Concluding Remarks

In vitro prion propagation methodologies supposed a breakthrough in the detection of minute amounts of prions in different tissues and body fluids, permitting for the first time to develop systems for ante-mortem TSE diagnosis based on the detection of the pathognomonic biomarker of disease, PrP^Sc^. Thus, the gold standard for definitive diagnosis of prion diseases (post-mortem detection of PrP^Sc^ in CNS) is being substituted by the analysis of lymphoid tissues or associated mucosa by rapid tests in animals [18] and the analysis of surrogate biomarkers or the detection of PrP^Sc^ through RT-QuIC in CSF in humans. Moreover, detection of minute amounts of PrP^Sc^ in different tissues and body fluids using in vitro prion propagation methods is a very active research field that is rapidly evolving, providing improved protocols that allow achieving unprecedented sensitivity and specificity [106].

However, although CSF-based RT-QuIC is already used in clinical practice and has been included as diagnostic criteria for sCJD in some countries, the tests based on PrP^Sc^ propagation in vitro need to be further tested for their generalized application. Multicentric studies with the same samples and using larger cohorts might be required to face standardization issues. Similarly, developing protocols to standardize sample collection and handling will be of great importance as has been already demonstrated in the case of surrogate biomarkers [76].

Undoubtedly, cell-free systems for prion amplification will continue to evolve and provide better results, and their implementation in clinical practice worldwide in a near future is foreseeable. Moreover, their adaptation to other protein misfolding-related neurodegenerative diseases may also revolutionize diagnosis of other, much more prevalent disorders, such as Parkinson’s disease [36]. Together, with other indicators, such as neuroimaging or surrogate biomarker analysis, detection of PrP^Sc^ in pre-clinical phases of the disease using prion propagation techniques has the potential to become the gold standard in TSE diagnosis, being based on a highly specific, pathognomonic biomarker.

## Figures and Tables

**Table 1 biomolecules-10-00469-t001:** List of studies performed using Real Time-Quaking Induced Conversion (RT-QuIC) in cerebrospinal fluid (CSF) from patients affected by different prion diseases. A summary of the most relevant parameters such as RT-QuIC substrate, assay sensitivity, and specificity are included. Creutzfeldt-Jakob disease (CJD), Sporadic Creutzfeldt-Jakob disease (sCJD), iatrogenic Creutzfeldt-Jakob disease (iCJD), prion protein (PrP), cerebrospinal fluid (CSF), Gerstmann-Sträussler-Scheinker syndrome (GSS), genetic Creutzfeldt-Jakob disease (gCJD), Familial Fatal Insomnia (FFI), Amyotrophic Lateral Sclerosis (ALS), Variably protease-sensitive prionopathy (VPSPr).

Prion Diseases	RT-QuIC Substrate	Sensitivity	Specificity	Observations	Reference
sCJD iCJD	Full-length recombinant human 129M PrP	>80%	100%	Just two iCJD cases. Controls included other neurodegenerative disorders (Alzheimer’s disease, Parkinson’s disease, etc.) All samples correspond to post-mortem collected CSF.	Atarashi et al. 2011 [92]
sCJD	Full-length recombinant Syrian hamster PrP	89%	99%	Controls included patients suspected of sCJD but finally diagnosed with other neurodegenerative disorders. All samples correspond to post-mortem collected CSF.	McGuire et al. 2012 [94]
GSS (P102L) gCJD (E200K, V203I) FFI	Full-length recombinant human 129M PrP	GSS 89% gCJD (E200K) 81.8% gCJD (V203I) 100% FFI 83.3%	100%	14-3-3 and tau analysis of the GSS and FFI samples detected only 20% and 8.3% of the positive cases, respectively.	Sano et al. 2013 [95]
sCJD gCJD	Full-length recombinant Syrian hamster PrP	70%	100%	CSF samples were obtained from patients with possible or probable CJD (alive) and with other neurologic disorders (Alzheimer’s disease, Parkinson’s disease, etc.)	Orrú et al. 2014 [96]
sCJD	Truncated recombinant Syrian hamster PrP (90–231)	96%	100%	CSF samples were obtained from patients with possible or probable CJD at the time of sampling, as well as from the patients with other neurologic disorders, including Alzheimer’s disease, ALS, atypical Parkinsonism, etc.	Orrú et al. 2015 [97]
sCJD gCJD (E200K,V210I) FFI	Sheep-Syrian hamster chimeric recombinant PrP (Syrian hamster 14–128 followed by sheep residues 141 to 234)	85%	99%	Control group composed by patients with either clinically or pathologically defined alternative diagnosis (Alzheimer’s disease, Lewy body dementia, Parkinson’s disease, psychiatric disorders, etc.) Lumbar puncture in sCJD samples was done in early, middle, or late disease stage.	Cramm et al. 2015 and 2016 [98,99]
sCJD	Full-length recombinant human 129M PrP	76.5%	100%	Negative control with artificial CSF.	Park et al. 2016 [100]
sCJD gCJD (E200K and V210I) GSS (P102L)	Truncated recombinant Syrian hamster PrP (90–231)	94%	100%	Controls included patients with other neurodegenerative diseases (multiple sclerosis, Alzheimer’s disease, etc.).	Groveman et al. 2017 [101]
sCJD VPSPr gCJD	Full-length recombinant Syrian hamster PrP	sCJD 75.9–82.7% VPSPr 0% gCJD 91.3%	99.4%	Two hundred and twenty-seven, 97, and 29 samples of definite, probable, and possible sCJD were analyzed; 348 cases of non-CJD patients were used as negative controls. Along with these, 1 case of VPSPr and 46 cases of gCJD were also tested.	Lattanzio et al. 2017 [72]
sCJD gCJD (E200K, V210I and V180I) GSS (P102L)	Truncated recombinant Syrian hamster PrP (90–231)	sCJD 95% gCJD 75%	100%	All non–prion disease control CSF samples, including those originally with suspected prion disease, were negative. The single case of gCJD with V180I mutation was always negative. Probable, possible, and suspected cases were included, being diagnosis confirmed in all cases post-mortem.	Bongianni et al. 2017 [102]
sCJD gCJD (E200K and V210I) FFI GSS (A117V, P102L)	Truncated Recombinant Syrian Hamster PrP (90–231)	95%	98.5%	Specificity is reduced due to a repeatedly positive Lewy Body Dementia case that may have also had a subclinical prion disease. The CSF RT-QuIC differentiated 94% of cases of sporadic Creutzfeldt-Jakob disease (sCJD) MM1 from the sCJD MM2 phenotype, and 80% of sCJD VV2 from sCJD VV1.	Foutz et al. 2017 [103]
sCJD gCJD (E200K and V210I) GSS (P102L) VPSPr	Truncated recombinant Syrian hamster PrP (90–231)	sCJD ranging from 90% to 100% depending on the subtype gCJD 100% GSS 25% VPSPr 100%	100%	CSF Analysis of 339 patients: 166 definite CJD 73 probable CJD 100 negative CJD	Franceschini et al. 2017 [104]
sCJD	Truncated recombinant Syrian hamster PrP (90–231)	96%	100%	Control cases included patients diagnosed by many other neurodegenerative disorders including Alzheimer’s disease, Lewy body dementia, Parkinson’s disease, etc.	Fiorini et al. 2020 [105]

**Table 2 biomolecules-10-00469-t002:** List of studies performed using different techniques for the detection of scrapie isoform of the prion protein (PrP^Sc^) in blood and blood fractions. The blood fraction in which PrP^Sc^ presence was checked is detailed, as well as the disease stage at the sampling time when available. Prion protein (PrP), Variant Creutzfeldt-Jakob disease (vCJD), Chronic Wasting Disease (CWD), bovine spongiform encephalopathy (BSE), Real Time Quaking Induced Conversion (RT-QuIC), Protein Misfolding Cyclic Amplification (PMCA), Surround Optical Fiber Immunoassay (SOFIA), Direct Detection Assay (DDA), Rocky Mountain Laboratory mouse-adapted scrapie strain (RML).

Assay	Species	Prion Strain	Blood Component	Disease Phase	Substrate	Sensitivity	Specificity	Reference
RT-QuIC	Human Hamster	vCJD 263K	Plasma Plasma & serum	Post-mortem and preclinical	Full-length recombinant Syrian hamster PrP Full-length recombinant human PrP Sheep–Syrian hamster chimeric recombinant PrP (Syrian hamster 14–128 followed by sheep residues 141 to 234)	100% 100%	100% 100%	Orrú et al. 2011 [252]
RT-QuIC	Cervid	CWD	Whole blood	Clinical and preclinical	Truncated recombinant Syrian hamster PrP (90–231)	>90%	100%	Elder et al. 2013 [253] Elder et al. 2015 [254]
PMCA	Hamster	263K	Buffy coat	Clinical Preclinical	Syrian hamster brain homogenate	89% 20–60%	100% 100%	Castilla et al. 2005 [21] Saá et al. 2006 [243]
PMCA	Sheep	Natural Scrapie infection	Buffy coat	Post-mortem	Sheep brain homogenate (V_136_R_154_Q_171_/V_136_R_154_Q_171_, A_136_R_154_Q_171_/A_136_R_154_Q_171_ and A_136_R_154_Q_171_/ V_136_R_154_Q_171_)	100%	100%	Thorne et al. 2008 [255]
PMCA	Sheep Macaque Human	Sheep-BSE vCJD vCJD	Buffy coat	Preclinical (Sheep-BSE & Macaque vCJD) Clinical (Human vCJD)	tgBov (Bovine PrP, line tg110), tga20 (murine PrP), tg338 (ovine V_136_R_154_Q_171_ PrP), tgShXI (ovine A_136_R_154_Q_171_ PrP variant) and tg650 (M_129_ variant of the human PrP) brain homogenates	100% 75%	100% 100%	Lacroux et al. 2014 [256]
PMCA	Macaque	vCJD	Whole blood	Preclinical and clinical	tgHu129M (M_129_ variant of the human PrP) brain homogenate	96–100%	100%	Concha-Marambio et al. 2020 [257]
PMCA	Cervid	CWD	Whole blood	Preclinical and clinical	tg1536 (mule deer PrP) brain homogenate	53–100%	100%	Kramm et al. 2017 [258]
Plasminogen bead-capture coupled to PMCA	Human	vCJD	Plasma	Post-mortem	tg338 (ovine V_136_R_154_Q_171_ PrP) and tg650 (M_129_ variant of the human PrP) brain homogenates	81.5–100%	96.5–100%	Bougard et al. 2016 [259]
Surround Optical Fiber Immunoassay (SOFIA) coupled to PMCA	SheepCervid	Natural and experimental Scrapie CWD	Plasma	Preclinical and clinical	Hamster, Sheep and deer brain homogenates	100% 100%	100% 100%	Rubenstein et al. 2010 [260]
Rapid ligand-based Immunoassay	Sheep	Natural Scrapie infection BSE	Buffy coat	Clinical and preclinical	-	33% (Preclinical Scrapie) 59–60% (Clinical Scrapie) 71% (Clinical BSE)	100%	Terry et al. 2009 [246]
Monoclonal antibody and streptavidin Immunoassay	Sheep and goat	Natural and experimental Scrapie	Whole blood	Post-mortem	-	100%	100%	Soutyrine et al. 2017 [261]
Raman spectroscopy	Sheep	Natural Scrapie	Membrane-rich fraction from Blood Cells	Post-mortem	-	100%	100%	Carmona et al. 2004 [210]
Infrared spectroscopy	Cattle	BSE	Serum	Clinical	-	>85%	>90%	Martin et al. 2004 [211]
In vitro amplification coupled to fluorescent amplification catalyzed by T7 RNA polymerase technique (Am-A-FACTTR)	Mouse Cervid	ME7 CWD	Plasma	Clinical and preclinical	Mouse and mule deer brain homogenates	100%	100%	Chang et al. 2007 [218]
Misfolded Protein Diagnostic	Sheep Mouse Human Squirrel monkey	Natural Scrapie Fukuoka-1 (GSS derived) SCJD (Human and monkey)	Plasma and serum	Clinical	-	100%	100%	Pan et al. 2007 [216]
Atomic Dielectric Resonance Spectroscopy	Human	vCJD sCJD	Whole blood	Clinical	-	100%	100%	Fagge et al. 2007 [217]
Fluorescence Intensity Distribution Analysis	Sheep	Scrapie	Plasma	Clinical	-	100%	100%	Bannach et al. 2012 [262]
Prototype blood-based vCJD assay	Human	vCJD	Plasma	Post-mortem	-	71.4%	100%	Jackson et al. 2014 [263]
Commercially available Amorfix EP-vCJD blood screening assay	Human	vCJD	Citrated plasma	-	-	97.6–99.9%	100%	Guntz et al. 2010 [249]
Direct immunodetection of Surface-Bound Material	Human	vCJD sCJD	Whole blood	Post-mortem	-	71.4%	100%	Edgeworth et al. 2011 [264]
Direct Detection Assay (DDA)	Mouse	RML	Whole blood	Clinical and preclinical	-	100%	100%	Sawyer et al. 2015 [265]

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
