# Peer review of "Detection of Pathognomonic Biomarker PrPSc and the Contribution of Cell Free-Amplification Techniques to the Diagnosis of Prion Diseases"

_biomolecules, 2020, doi:10.3390/biom10030469_

Round 1

Reviewer 1 Report

This manuscript is the review paper on the detection of biomarker PrPSc using PMCA and RT-QuIC techniques. This review paper is so interesting. However, there are some minor points that the authors may consider.

Minor Comments

  1. Table 1 title, “List of studies performed using RT-QuIC and CSF from patients affected by different prion diseases” -> “List of studies performed using RT-QuIC in the CSF from patients affected by different prion diseases”

  1. In table 1, the order of “the reference” should be placed after the observations as table 2

  1. Table 2 title, “List of studies performed using different techniques for the detection of prions in blood and blood fractions” -> “List of studies performed using different techniques for the detection of PrPSc in blood and blood fractions”

  1. In Table 2, Page 15, Second column

         236K -> 263K

  1. In Abstract, there are no abbreviations of RT-QuIC and PMC

  1. Table 2, the substrates of RT-QuIC and PMCA should be inserted.

  1. Page 2, line 92, immunohistochemistry  or  Page 10, line 389, IHC

Author Response

This manuscript is the review paper on the detection of biomarker PrPSc using PMCA and RT-QuIC techniques. This review paper is so interesting. However, there are some minor points that the authors may consider.

Minor Comments

  1. Table 1 title, “List of studies performed using RT-QuIC and CSF from patients affected by different prion diseases” -> “List of studies performed using RT-QuIC in the CSF from patients affected by different prion diseases”

 We appreciate the correction. Table 1 title has been changed accordingly.

  1. In table 1, the order of “the reference” should be placed after the observations as table 2

We thank the reviewers’ recommendation. As indicated, the column with the references has been placed after the Observations column.

  1. Table 2 title, “List of studies performed using different techniques for the detection of prions in blood and blood fractions” -> “List of studies performed using different techniques for the detection of PrPSc in blood and blood fractions”

 As suggested, title of Table 2 has been modified.

  1. In Table 2, Page 15, Second column

         236K -> 263K

We apologize for the mistake, it has been corrected. 

  1. In Abstract, there are no abbreviations of RT-QuIC and PMC

Following the reviewers’ recommendation both abbreviations have been included in the Abstract.

  1. Table 2, the substrates of RT-QuIC and PMCA should be inserted.

We agree with the reviewer and think the information about substrates will be of interest. Thus, a new column has been added to table 2 to include the substrates used in PMCA and RT-QuIC assays.

  1. Page 2, line 92, immunohistochemistry  or  Page 10, line 389, IHC

As suggested by the reviewer immunohistochemistry has been first defined as IHC (page 2 line 92) and then has been substituted throughout the text by IHC.

Reviewer 2 Report

See attached Word document

Author Response

Erana et al. provide an exhaustive, authoritative and timely review on a subject that is of utmost significance to the field. For the most part, the content is great, and well explained. However, there are many places where minor edits of the English are needed are/or would be clarifying. The readability of the text would be improved in many places by dividing very long, complicated sentences into two.

In general, there are multiple marathon paragraphs that would be more readable if divided up into subtopics.

We appreciate the positive comments from the reviewer and specially the recommendations regarding mistakes of the English. We also want to thank the reviewer for the general suggestion of making phrases shorter and subdividing paragraphs to improve the readability of the manuscript. Apart from the corrections directly indicated by the reviewer below (and for which we are really grateful), we have thoroughly reviewed the manuscript dividing as many long sentences as possible and new subheadings have been included in many sections in order to divide them in different subtopics. Please see changes performed in blue throughout the manuscript.

L21: Although would be a better word than "despite" here, and in many subsequent places in the manuscript. These two words have somewhat different meanings, and usually ‘although’ seems to have been meant in many contexts in which ‘despite’ was used.

We thank the reviewer the correction and the tip regarding the connectors although and despite. Several “Despite” has been accordingly substituted by “Although”, apart from the ones indicated in the corrections below. Again, we are grateful for the corrections that can be seen in blue throughout the manuscript.

L83: Ref #13 is not a good reference for this statement. Seeding is not proposed as a mechanism in this paper. In fact, for years after this paper, the author strongly objected to seeded polymerization of PrP as the mechanism of PrPSc propagation, preferring instead a heterodimer model and describing prion fibrils as being primarily in vitro artifacts. It was primarily Lansbury, Gadjusek and Caughey who initially pushed, and provided evidence for, seeding mechanisms for PrPSc formation that were initially envisioned by JS Griffith.

We apologize for the mistake. As pointed by the reviewer, ref. 13 has been substituted for the following two references:

  1. Kocisko DA, Come JH, Priola SA, Chesebro B, Raymond GJ, Lansbury PT, et al. Cell-free formation of protease-resistant prion protein. Nature. 1994;370(6489):471-4. doi: 10.1038/370471a0.
  2. Lansbury PT, Jr., Caughey B. The double life of the prion protein. Curr Biol. 1996;6(8):914-6. doi: S0960-9822(02)00624-3.

L87: Replace "spans and multiplying exponentially" with "spreads and multiplies exponentially".

We thank the correction. It has been modified accordingly.

L97: Despite --> Although

Corrected.

L153-154: Also in European diagnostic criteria used by the UK National CJD Research and Surveillance Unit (https://www.cjd.ed.ac.uk/sites/default/files/criteria_0.pdf).

The reviewer is right on pointing this out and thus, we have included the following comment in line 154: “Similarly, it was also included in European diagnostic criteria used by the UK National CJD Research and Surveillance Unit.”

L220: withdrawn à drawn

Corrected.

L229: that à though

Corrected.

L244-249: very long awkward sentence

We agree with the reviewer that the sentence was too long and awkward. In part it might be because we tried to include too much unnecessary information in the sentence. Therefore, for the sake of clarity, we have decided to eliminate the unnecessary information (mechanistic explanation of PrP level reduction in CSF). The sentence has been re-written to the following:

“In fact, total PrP levels were reported to be reduced in CSF of patients suffering from different neurodegenerative disorders, including CJD [73].”

L295: On the contrary à In comparison

Corrected.

L300: 2006 à 2011

We apologize for the mistake, it has been corrected.

L310: Despite àAlthough

Corrected.

L314: fix spelling of synucleinopathies

Corrected.

L321: the à that; insert ‘being’ after ‘while.

We appreciate the correction, it has been modified as indicated.

L324: Despite à Although

Corrected.

L328: what à that

Corrected.

L334: what à that

Corrected.

L361: delete "what has been"

Corrected.

L371: insert ‘the’ before ‘preclinical’

Corrected.

L372: in à within

Corrected.

L377: Despite àAlthough

Corrected.

L379: insert ‘with it’ in front of ‘being’

Corrected.

L380: offering enhancedàto enhance

Corrected.

L383-387: long awkward sentence

We agree with the reviewer that the sentence is difficult to follow. It has been re-written as follows:

“The original prion in mink shows early affectation of lymphoid organs by intracerebral inoculation. Upon oral inoculation, PrPSc is detected even earlier in retropharyngeal and mesenteric lymph nodes, then in spleen and finally in GALT. Moreover, in contrast to intracerebrally inoculated animals, PrPSc was found in rectal mucosa-associated lymphoid tissue (RAMALT) for orally inoculated minks [51].”

L417: sensitive à sensitivity

Corrected.

L420: through intracerebral à intracerebrally

Corrected.

L435: delete ‘methods’

Corrected.

L447: insert ‘namely’ in front of ‘the RAMALT’

Corrected.

L459: "being found mainly on those affected by Kuru that showed" --> with those affected by Kuru showing

Corrected.

L465: delete ‘already’

Deleted.

L470: delete "presence of"

Deleted.

L471: "other prion disease, neither iCJD," --> iCJD,

Corrected.

L483: despite à although

Corrected.

L513: insert ‘a’ after ‘thus’

Corrected.

L525: insert ‘a’ after ‘or’

Corrected.

L526: insert ‘and found to be’ after ‘analyzed,’

Corrected.

L535-537: This statement is not really true. As the authors have explained, the rapid tests are helpful but not as sensitive as they could/should be. RT-QuIC and PMCA have been demonstrated to significantly enhance diagnostic sensitivity for CWD in cervids.

We agree with the reviewer that this sentence is an overstatement, so the sentence has been modified to include the idea that in vitro amplification methods enhance diagnostic sensitivity also in CWD and scrapie cases with respect to rapid tests and IHC.

“Since rapid tests based on ELISA offer good specificity and sensitivity for these cases, cell-free prion propagation techniques have not been extensively used, despite showing enhanced sensitivity compared to the previous methods.”

L562: Meaning of ‘10E-5 lower’ is not as seems to be intended. Should say 10E5-fold lower.

Corrected.

L576: insert ‘this situation’ after ‘complicated’

Corrected.

L592: delete ‘presence’

Corrected.

L594: insert ‘an’ before ‘event’

Corrected.

L600: what àthat

Corrected.

L633: insert ‘off’ after ‘far’

Corrected.

L701: up to --> as little as

Corrected.

L738: on àof

Corrected.

L746: aiming its --> aimed at using PrPSc

Corrected.

L750: what à that

Corrected.

L770: hamster àhamsters

Corrected.

L771: albeit à although

Corrected.

L387-341: long awkward sentence

In order to improve clarity the sentence has been divided in two and re-written as follows:

“Importantly, it has also been applied to determinate the time course of prion excretion in urine of three different orally CWD infected-cervid species at the preclinical stage: elk, mule deer and white-tailed deer. They found that prion excretion started as early as 6 months post-infection in feces whereas their detection in urine resulted 10 times less frequent, finding PrPSc at 6 months only in one out of two white-tailed deer and requiring 18 months to be detected in one of the two elks and in the only mule deer tested [291].”

L846: insert ‘were’ after ‘problems’

Corrected.

L861: loss àlose

Corrected.

L870: on à in

Corrected.

L880: 1.2·107: The exponent should be negative here.

The reviewer is right, we apologize for the mistake. It has been corrected.

L921: A study that is highly relevant to this paragraph but is not mentioned: Orru et al., Prion seeds distribute throughout the eyes of sporadic Creutzfeldt-Jakob disease patients. mBio 2018 9:e02095-18.

We would like to apologize for the absence of such a relevant study in this section. A new sentence has been added to introduce the reference (#317 in the reviewed version) and the main findings from the study:

“In fact, a study performed using RT-QuIC in post-mortem sCJD patients’ samples, revealed PrPSc in cornea, lens, ocular fluid, retina, choroid, sclera, optic nerve, and extraocular muscle [317].”

Reviewer 3 Report

Major comments:

This is a very thorough review. I think it would benefit from the following:

  • English editing, a lot of the terminology is incorrect and the sentences confusing  
  • Brevity, the are several repetitions and sentences that could be shortened/ split for clarity
  • Figures and/schematics. The tables are great, but I would add maybe a figure or two to summarize which prion strain was detected, in which tissue and using which technique. 

Minor comments:

Lines 158-160 This sentence is confusing and misleading. The RT-QuIC has been reported to be able to detect vCJD prion seeding activity in brain and skin samples, this information is in references 42 and 323.

I think this review would benefit from a little bit more detail when navigating the multitude of prion strains. Just as an example, lines 200-201: what do the authors mean by “most relevant prion strains”

The tables have some formatting issues that should be addressed

Author Response

Reviewer 3

Major comments:

This is a very thorough review. I think it would benefit from the following:

  • English editing, a lot of the terminology is incorrect and the sentences confusing  

We agree with the reviewer on the fact that language needs to be improved throughout the article. For that we have done a thorough revision trying to improve language and eliminate or fix confusing sentences. Together with the corrections form other reviewers, we hope the language in the manuscript has been substantially improved. Changes can be seen in blue throughout the manuscript.

  • Brevity, the are several repetitions and sentences that could be shortened/ split for clarity

We appreciate the comment from the reviewer. We have tried to split long sentences and eliminate duplications in order to improve clarity. Some redundancies have been also removed for the sake of clarity. We expect that the changes done through the text are enough to improve its readability. See changes in the text in blue.

  • Figures and/schematics. The tables are great, but I would add maybe a figure or two to summarize which prion strain was detected, in which tissue and using which technique. 

We are glad that the reviewer considers the tables useful and we have really thought about the proposal of including some figure summarizing part of the review. However, we have found it impossible to do accurately, since the information available about strains is in many of the articles scarce. Many of them talk about the strains generally (no information about CWD types, sCJD subtypes, the different gCJD cases used…). Therefore, we considered that a figure with such information would not add much with respect to the information already included in the text. Moreover, tables 1 and 2 already include the most relevant information on the most interesting and widely used body fluids (CSF and blood), detection techniques and prion strains used as proof of concept. In addition, the tables have been completed attending to the comments from another reviewer and when possible, strain details have been added in the text following a posterior suggestion. Thus, we hope that enough detail has been added in the text and in the two tables to make clear in which tissues and fluids prions have been detected and by which means.

Minor comments:

Lines 158-160 This sentence is confusing and misleading. The RT-QuIC has been reported to be able to detect vCJD prion seeding activity in brain and skin samples, this information is in references 42 and 323.

The reviewer is right and we apologize for the mistake. The sentence made reference to 1st generation RT-QuIC, but it is true that using recombinant bank vole protein as a substrate vCJD detection problems were solved as well as lack of detection of many atypical prion diseases. Therefore, to reflect this, few sentences in lines 158-164 have been re-written:

“The main problem delaying its generalized use in clinical practice is that first-generation RT-QuIC, performed using recombinant hamster PrP, was unable to detect some specific prions (i.e. variant CJD, the human prion disease acquired from Bovine spongiform encephalopathy contaminated sources; some GSS cases due to specific point mutations; etc.) and sensitivity was quite low for other subtypes [19]. Nonetheless, the use of a new recombinant PrP as substrate [42] has provided a solution to this problem, likely making the detection of PrPSc in body fluids or tissues by RT-QuIC the next gold standard for ante-mortem diagnosis of TSE”

I think this review would benefit from a little bit more detail when navigating the multitude of prion strains. Just as an example, lines 200-201: what do the authors mean by “most relevant prion strains”

We agree with the reviewer that in some points of the manuscript it is useful to go into more detail to describe the strains. For that, we have added details in some sentences such as the one in line 200, 287 or line 296. In some cases, the information about strains could not be added because were not described in detail in the original reference.  However, since it is not the main goal of the manuscript, the rest of the strain descriptions have been considered enough, although further detail has been added when possible.

The tables have some formatting issues that should be addressed

We agree with the reviewer on this point, but table formatting is done by the journal in this case. However we will be vigilant to avoid any formatting issue in the final version after edition by the journal.
